# Beyond the Proxy: Trajectory-Distilled Guidance for Offline GFlowNet Training

Ruishuo Chen [1]   Xun Wang [1]   Rui Hu [1]   Zhuoran Li [1]   Longbo Huang [1]

## Abstract

Generative Flow Networks (GFlowNets) excel at sampling diverse, high-reward objects. In many practical applications where active reward queries are infeasible, these models must be trained using static offline datasets. Prevailing training methods typically rely on a proxy model to provide reward feedback for online sampled trajectories. However, constructing a reliable proxy is often challenging due to data scarcity or high evaluation costs. While existing proxy-free approaches attempt to address this, they often impose coarse constraints that limit the model's ability to explore effectively. To overcome these limitations, we propose **Trajectory-Distilled GFlowNet (TD-GFN)**, a novel proxy-free training framework. TD-GFN utilizes inverse reinforcement learning (IRL) to extract dense, transition-level edge rewards from offline trajectories, providing rich structural guidance for efficient exploration. Crucially, to ensure robustness, these rewards guide the policy indirectly through DAG pruning and prioritized backward sampling. This design ensures that gradient updates rely exclusively on ground-truth terminal rewards from the dataset, thereby preventing error propagation. Empirical results demonstrate that TD-GFN significantly outperforms a broad range of existing baselines in both convergence speed and sample quality, establishing a more robust and efficient paradigm for offline GFlowNet training.

## 1. Introduction

Generative Flow Networks (GFlowNets or GFNs) (Bengio et al., 2021; 2023) are a powerful class of generative models designed to sample compositional objects in proportion to their rewards. They have demonstrated significant success across diverse domains, including molecule discovery (Bengio et al., 2021), biological sequence design (Jain et al., 2022), combinatorial optimization (Zhang et al., 2025a), and generative model fine-tuning (Hu et al., 2024a; Liu et al., 2025). Despite this success, active environment interaction for reward feedback is often impractical in practical applications due to expensive experiments or the need for human evaluation. This constraint necessitates training GFlowNets from pre-collected offline datasets.

The standard paradigm involves training a proxy model to approximate reward signals from the dataset, which the GFlowNet then queries to facilitate policy optimization. However, constructing a reliable reward model is resource-intensive, typically requiring large, diverse datasets and significant domain expertise (Ouyang et al., 2022; Jain et al., 2023a). Furthermore, reward queries targeting out-of-distribution samples, or states inaccurately estimated relative to the proxy, can propagate errors through the gradients and compromise the learned policy. These challenges are especially acute in domains such as language modeling (Ziegler et al., 2019; Dalrymple et al., 2024), recommendation systems (Chen et al., 2024), and autonomous driving (Xie et al., 2024), where obtaining ground-truth rewards is prohibitively expensive because of the high costs associated with expert annotation or experimental trials.

Conversely, execution trajectories in these domains are inherently generated and readily available. While often overlooked in proxy-based methods, this trajectory-level information is a cornerstone of offline reinforcement learning (Kumar et al., 2020; Kostrikov et al., 2022) that has yet to be fully exploited for GFlowNets. As recent works expand GFlowNets into a broader spectrum of domains (Liu et al., 2023; Hu et al., 2024a; Zhu et al., 2025) to enable diverse generation, a critical question emerges: how can we effectively leverage these offline trajectories to train GFlowNets when the available terminal rewards are insufficient to construct a reliable proxy model?

Recent studies, such as RO-GFlowNets (Wang et al., 2023) and COFlowNet (Zhang et al., 2025b), have explored learning directly from offline trajectories to eliminate dependence on proxy models. However, these approaches typically impose coarse constraints to align the policy with the dataset. Such practices can restrict generalization, inhibit effective

---

[1]Institute for Interdisciplinary Information Sciences, Tsinghua University, Beijing, China. Correspondence to: Longbo Huang <longbohuang@tsinghua.edu.cn>.

*Proceedings of the 43rd International Conference on Machine Learning*, Seoul, South Korea. PMLR 306, 2026. Copyright 2026 by the author(s).

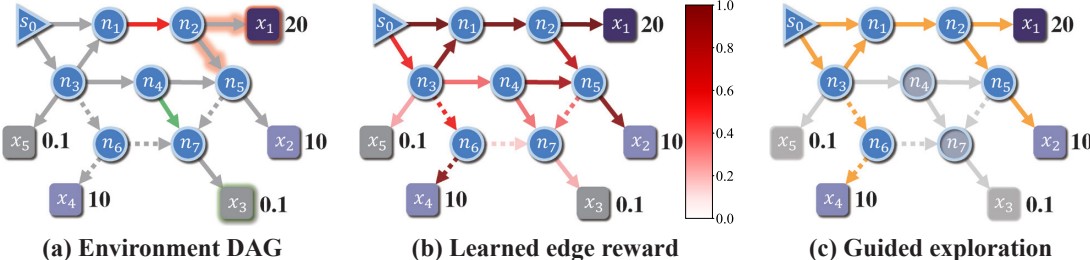

**(a) Environment DAG**  **(b) Learned edge reward**  **(c) Guided exploration**

*Figure 1.* Illustration of our motivation. All subfigures depict an environment DAG, where solid edges represent transitions observed in the dataset and dashed edges indicate unobserved transitions. Reward values are annotated next to the terminal states. (a) Different edges contribute unequally to the final learned distribution. (b) Edge contributions as inferred by TD-GFN. (c) Selective exploration guided by the learned edge contributions.

exploration, and heighten sensitivity to data quality.

Rather than imposing such naive constraints, we introduce **Trajectory-Distilled GFlowNet (TD-GFN)**, a novel *proxy-free* framework that extracts rich structural information from the environment's directed acyclic graph (DAG) to guide its policy. Our work is based on an insight that different edges in the DAG contribute unequally to learning an effective GFlowNet policy, an observation also shared by prior work (Silva et al., 2025) in different contexts. For example, as shown in Figure 1(a), removing edge $(n_1 \rightarrow n_2)$ blocks access to a terminal node with reward 20, whereas removing edge $(n_4 \rightarrow n_7)$ only affects a low-reward node (0.1). By leveraging the theoretical equivalence between GFlowNet training and reinforcement learning (Tiapkin et al., 2024), we employ inverse reinforcement learning (IRL) on a re-balanced offline dataset to quantify this varying edge importance. This approach allows us to distill transition-level scores termed edge rewards, as illustrated in Figure 1(b). These learned rewards provide dense, intermediate guidance that enables the policy to generalize across the DAG, fundamentally differing from proxy models in both learning objective and application.

To ensure robustness, we utilize these edge rewards in an *indirect* yet effective manner. First, we prune low-utility transitions based on their edge rewards, yielding a more compact and expressive action space that prioritizes high-value trajectories (Figure 1(c)). Second, we introduce a prioritized backward sampling procedure guided by both terminal and edge rewards, which strategically allocates the model's attention across terminal states and intermediate pathways during training. Throughout this process, gradient-based updates to our policy rely exclusively on ground-truth terminal rewards from the dataset. This indirect application insulates the policy from potential inaccuracies in the IRL phase, preventing the error propagation that plagues proxy-based methods. More importantly, the dense structural guidance provided during decision-making enables TD-GFN to explore high-reward samples with superior efficiency.

Extensive experiments confirm these advantages, demonstrating that our proposed TD-GFN trains with superior efficiency and reliability, significantly outperforming a wide range of baselines in convergence speed and final sample quality. These results establish TD-GFN as a more robust, stable, and efficient paradigm for offline GFlowNet training, marking a significant step forward from conventional methods that unlocks the potential for deploying GFlowNets in a broader spectrum of complex domains and scenarios.

In summary, our key contributions are as follows:

- **A Novel Proxy-Free Paradigm:** We introduce TD-GFN, a paradigm for training GFlowNets from offline trajectory data that eliminates dependence on proxy models and avoids out-of-distribution reward queries. By employing IRL to distill transition-level edge rewards from a rebalanced dataset, we provide structured guidance that to guide the policy's training and generalization across the environment.

- **Robust Guidance Mechanism:** We propose two mechanisms for indirect policy guidance using learned edge rewards. Through DAG pruning and prioritized backward sampling, TD-GFN facilitates efficient training under dense guidance while ensuring that gradient updates depend exclusively on ground-truth rewards, thereby preventing error propagation.

- **State-of-the-Art Performance and Efficiency:** Through extensive experiments, we demonstrate that TD-GFN establishes a new state-of-the-art for offline GFlowNet training. Our method significantly outperforms diverse baselines by achieving a superior fit to the target distribution, faster convergence, and higher sample quality across multiple benchmarks.

**Code Availability.** Source code and scripts to reproduce all experiments are available at https://github.com/Chenruishuo/TD-GFN.

## 2. Preliminaries

In this section, we describe Generative Flow Networks (GFlowNets or GFNs) and the maximum causal entropy inverse reinforcement learning (IRL) framework, which help clarify our proposed method. Additional related works are provided in Appendix C for completeness.

### 2.1. GFlowNets

GFlowNets (Bengio et al., 2021; 2023) are formulated on an environment structured as a directed acyclic graph (DAG), denoted by $G = (V, E)$, where $V$ and $E$ represent the sets of nodes and directed edges, respectively. A node $s'$ is defined as a child of $s$ if there exists a directed edge $(s \rightarrow s') \in E$; in this case, $s$ is considered a parent of $s'$. The graph comprises a unique root node $s_0 \in V$ with no incoming edges and a subset of terminal nodes $\mathcal{X} \subseteq V$ with no outgoing edges.

A positive reward function $R(x)$ is defined over the terminal nodes $\mathcal{X}$. GFlowNets aim to learn a stochastic forward policy $\mathcal{P}_F(s'|s)$ such that the probability of reaching any terminal node $x$ is proportional to its reward $R(x)$. This objective is achieved by generating sequences of transitions where the agent selects a child node at each step. We assume deterministic environment dynamics where each action uniquely determines the next state by traversing a specific DAG edge, following standard conventions in GFlowNet literature (Bengio et al., 2021; Jain et al., 2022; Bengio et al., 2023).

Recent work (Tiapkin et al., 2024) establishes a formal equivalence between GFlowNet training and entropy-regularized RL under specific conditions: a discount factor $\gamma = 1$, an entropy regularization coefficient $\lambda = 1$, and a modified reward function. This reward is determined by a fixed backward policy $\mathcal{P}_B(s|s')$ that samples parent nodes in the DAG in reverse from a child node:

$$\widetilde{R}(s, s') = \begin{cases} \log \mathcal{P}_B(s|s'), & \text{if } s \notin \mathcal{X} \cup \{s_f\}, \\ \log R(s), & \text{if } s \in \mathcal{X}, \\ 0, & \text{if } s = s_f, \end{cases} \quad (1)$$

where $s_f$ denotes an additional absorbing state. Consequently, learning the forward policy reduces to solving the following entropy-regularized policy optimization problem:

$$\mathcal{P}_F^* = \arg\max_{\pi} \left( \lambda \mathcal{H}(\pi) + \mathbb{E}_{\pi} \left[ \sum_{t=0}^{\infty} \gamma^t \widetilde{R}(s_t, s_{t+1}) \right] \right), \quad (2)$$

where $\mathcal{H}(\pi) = \mathbb{E}_{\pi} \left[ -\log \pi(s'|s) \right]$ denotes the causal entropy of the policy. The expectation $\mathbb{E}_{\pi}$ is taken with respect to trajectories generated by $s_0 \sim P_0$ and $s_{t+1} \sim \pi(\cdot|s_t)$, where $P_0$ is the initial state distribution that corresponds to a Dirac delta distribution in the GFlowNet setting.

In this paper, we address the problem of training a GFlowNet using an offline dataset $\mathcal{D} = \{\tau_i = (s_0, s_1, \ldots, s_{T_i}, R(s_{T_i}))\}_{i=1}^{M}$, which consists of trajectories of constructed objects generated by an unknown behavior policy, along with their corresponding ground-truth rewards.

### 2.2. Maximum causal entropy IRL

Maximum causal entropy inverse reinforcement learning (Ziebart et al., 2008; 2010) aims to recover a reward function $r(s, s')$ that both characterizes expert behavior under a policy $\pi_E$ and maximizes the causal entropy of the learned policy. This objective is formalized as follows:

$$\begin{aligned} \min_{r} \quad & \left( \max_{\pi} \left( \lambda \mathcal{H}(\pi) + \mathbb{E}_{\pi} \left[ \sum_{t=0}^{\infty} \gamma^t r(s_t, s_{t+1}) \right] \right) \right. \\ & \left. - \mathbb{E}_{\pi_E} \left[ \sum_{t=0}^{\infty} \gamma^t r(s_t, s_{t+1}) \right] \right), \end{aligned} \quad (3)$$

where $\mathcal{H}(\pi)$ denotes the causal entropy of policy $\pi$, consistent with the definition in Equation (2). In our framework, we adapt this objective to infer an edge-level reward function $R_E : E \rightarrow \mathbb{R}$ that reflects the expert policy's preferences for specific transitions within the graph.

## 3. Method

In this section, we introduce **Trajectory-Distilled GFlowNet (TD-GFN)**, a novel proxy-free training framework. In contrast to existing methods that restrict the policy to the dataset distribution (Wang et al., 2023; Zhang et al., 2025b), TD-GFN facilitates a superior balance between exploration and exploitation by strategically allocating attention across the environment DAG.

The training pipeline is illustrated in Figure 2. Initially, TD-GFN extracts an *edge reward function* from a rebalanced dataset using inverse reinforcement learning (IRL). This function captures the relative importance of transitions, providing principled guidance for policy optimization. Using these rewards, TD-GFN prunes the DAG to eliminate low-utility transitions and optimizes the policy through prioritized backward sampling on the resulting subgraph. This design enables effective generalization to unseen regions of the state space while maintaining alignment with high-reward behaviors. The following subsections detail each component, and a complete summary of the procedure is provided in Algorithm 1 (Appendix A).

### 3.1. Edge rewards extraction via IRL

Since different edges in the environment DAG contribute unequally to policy learning, we aim to quantify this disparity to enable more targeted and generalizable training. However, measuring the contribution of individual transitions under

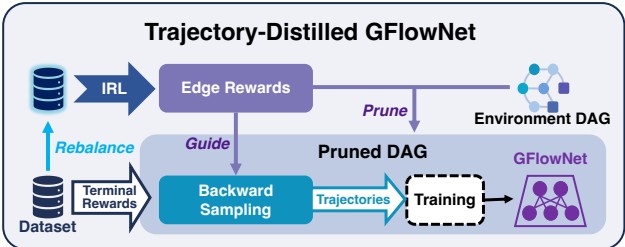

*Figure 2.* An overview of the complete training pipeline of the TD-GFN framework.

stochastic GFlowNet policies remains challenging. To address this, we leverage the theoretical equivalence between GFlowNet training and entropy-regularized reinforcement learning (Equation (2)). Specifically, we apply the maximum causal entropy IRL framework (Ziebart et al., 2008; 2010) (Equation (3)) to extract a fine-grained edge-level reward function from offline trajectories. These learned edge rewards act as structured auxiliary signals that capture the relative importance of transitions, guiding the policy to generalize effectively across the environment DAG.

A primary challenge in applying IRL to offline data is that the underlying behavior policy may not represent expert behavior. To address this, we introduce a rebalancing strategy inspired by Hong et al. (2023) that biases trajectory sampling toward high-reward terminal states. Specifically, when estimating expectations under the expert policy $\mathbb{E}_{\pi_E}$, we sample trajectories $\tau = (s_0, s_1, \ldots, s_T) \in \mathcal{D}$ with probability $P(\tau) \propto R(s_T)$. This approach constructs a *rebalanced dataset* that approximates the visitation distribution of an expert policy. By adjusting edge visitation frequencies to favor high-reward regions, this reweighting scheme effectively creates a pseudo-expert dataset and aligns the learned edge rewards more closely with expert behavior.

Building on the well-known maximum causal entropy IRL framework GAIL (Ho & Ermon, 2016) which has been theoretically and empirically shown to effectively learn reward functions with strong generalization (Xu et al., 2020; Luo et al., 2022; 2024), we adopt the following adversarial training objective based on the rebalanced dataset:

$$\min_{\phi} \max_{\psi} \mathcal{L}(\psi, \phi) = \mathbb{E}_{s \sim \widetilde{\mathcal{D}}, s' \sim \pi_{\psi}} [\log D_{\phi}(s, s')]$$
$$+ \mathbb{E}_{(s,s') \sim \widetilde{\mathcal{D}}} [\log(1 - D_{\phi}(s, s'))] + \lambda \mathcal{H}(\pi_{\psi}), \tag{4}$$

where $\widetilde{\mathcal{D}}$ represents the distributions of nodes and edges (with slight abuse of notation), while $\pi_{\psi}$ is a parameterized policy and $D_{\phi} : E \to (0, 1)$ is a discriminative classifier. Both components are optimized iteratively.

However, the original GAIL framework induces a strictly non-negative reward function, rendering it incapable of recovering the true underlying reward in our setting, which may include negative values as shown in Equation (1). To

address this, we follow Kostrikov et al. (2019) and extract an unbiased edge-level reward function $R_E$ from the classifier:

$$R_E(s, s') = \log D_{\phi}(s, s') - \log (1 - D_{\phi}(s, s')) \tag{5}$$

Intuitively, the learned edge reward reflects the preference of the near-expert behavior policy implied by the rebalanced dataset, thereby quantifying the importance of each transition for learning a high-quality GFlowNet policy. This approach differs fundamentally from proxy reward models because it is not designed to predict terminal rewards. Instead, it provides indirect, non-gradient guidance to the policy, as detailed in the following section. The complete procedure for edge reward extraction is summarized in Algorithm 2 (Appendix A).

To understand what $R_E$ recovers in the population limit, we analyze Equation (3) under standard finite-horizon GFlowNet conventions (Appendix B). Let $\widetilde{R}(x) := w(x)R(x)$ denote the terminal weight of the rebalanced expert. By Proposition B.1, this rebalanced expert shares the backward policy $\mathcal{P}_B^*$ of the original GFlowNet, so the canonical edge reward $r^*(s, s') := \log \mathcal{P}_B^*(s \mid s')$ on $E$ is invariant to rebalancing.

**Theorem 3.1** (Unique Recovery of the Backward Policy).
*Restrict the population IRL problem in Equation (3) to rewards $r$ on $\bar{E}$ satisfying the terminal-boundary condition $r(x, s_f) = \log \widetilde{R}(x)$ for every $x \in \mathcal{X}$. Among all global minimizers, the unique reward also satisfying the parent-normalization gauge*

$$\sum_{p \in \mathrm{Pa}(s')} \exp(r(p, s')) = 1 \qquad \forall s' \in V \setminus \{s_0\}$$

*is the canonical edge reward $r^*$. Since $r^*$ is itself parent-normalized, the parent softmax in Equation (7) applied to $r^*$ recovers $\mathcal{P}_B^*$ exactly. More generally, if the learned $R_E$ satisfies $\|R_E - r^*\|_{\infty} \leq \varepsilon$, the recovered backward policy $\hat{\mathcal{P}}_B$ from Equation (7) obeys*

$$\sup_{s' \in V \setminus \{s_0\}} \mathrm{TV}(\hat{\mathcal{P}}_B(\cdot \mid s'), \mathcal{P}_B^*(\cdot \mid s')) \leq \tfrac{1}{2}(e^{2\varepsilon} - 1).$$

*Proofs appear in Appendix B (Theorem B.5 and Theorem B.7).*

Theorem 3.1 shows that $R_E$ targets the log conditional incoming-flow score and gives a quantitative recovery guarantee for the induced backward policy. Empirically, this recovery also generalizes beyond the observed transitions. As illustrated in Figure 1(b) and Appendix E.4, the learned edge rewards demonstrate strong generalization by assigning meaningful values to transitions absent from the dataset. This property encourages the policy to distribute attention throughout the environment DAG instead of remaining restricted to regions near the training data, a limitation common in prior methods (Wang et al., 2023; Zhang et al.,

2025b). This generalization capability is essential for the GFlowNet objective because accurately matching the target distribution requires the policy to navigate and evaluate states not explicitly represented in the offline data. Notably, this global edge-level guidance aligns with the core GFlowNet philosophy of using structured, multi-step decision-making to decompose complex sampling problems instead of attempting to model the distribution directly.

### 3.2. Reward-guided pruning and prioritized backward sampling

Although edge rewards provide meaningful transition-level signals, using them to shape policy gradients directly increases sensitivity to function approximation errors. This risk mirrors the error propagation issues observed with proxy rewards. To improve both robustness and efficiency, we introduce two mechanisms to shape the GFlowNet policy as illustrated in Figure 2. Specifically, we perform *reward-guided pruning* on the environment DAG and subsequently train the model using trajectories generated through *prioritized backward sampling* on the resulting pruned graph.

Guided by the learned edge rewards, TD-GFN removes low-utility edges from the environment DAG $G = (V, E)$. We approximate the distribution of edge rewards using a batch $\mathcal{D}_{R_E} = \{R_E(s_i, \text{selected}(\pi_\psi(\cdot \mid s_i)))\}_{i=1}^{\mathcal{B}}$, where $\pi_\psi$ is the imitation policy from Algorithm 2 serving as an expert surrogate. A transition $(s \to s')$ is pruned from the graph if it lies in the low-density region of this reward distribution:

$$R_E(s, s') < \text{mean}(\mathcal{D}_{R_E}) - K \cdot \text{std}(\mathcal{D}_{R_E}), \quad (6)$$

where $K$ is a pruning threshold hyperparameter and $\text{mean}(\cdot)$ and $\text{std}(\cdot)$ denote the empirical mean and standard deviation over the batch $\mathcal{D}_{R_E}$. This threshold-based strategy relies on the model's ability to distinguish low-reward edges from potentially useful ones. This approach protects the policy from numerical errors in the IRL phase and reduces the risk of performance degradation. We provide a sensitivity analysis for $K$ in Appendix E.5 and discuss alternative soft guidance mechanisms in Appendix F.

To analyze the robustness of this thresholding rule, we work with a path-based criterion. For each terminal $x \in \mathcal{X}$, define the *bottleneck score*

$$\beta(x) := \max_{P \in \mathcal{P}(x)} \min_{e \in P} r^*(e),$$

where $\mathcal{P}(x)$ denotes the set of directed paths from $s_0$ to $x$. The bottleneck score is the largest minimum edge score along any root-to-$x$ path under $r^*$; high $\beta(x)$ means $x$ is reachable through an edge-score margin that absorbs estimation noise.

**Theorem 3.2** (Robust Preservation and Distortion)**.** *Let $R_E$ be a learned edge reward with sup-norm error $\|R_E -$*

*$r^*\|_\infty \leq \varepsilon$, and let $\tau \in \mathbb{R}$ be any pruning threshold (Equation (6) instantiates $\tau = \text{mean}(\mathcal{D}_{R_E}) - K \cdot \text{std}(\mathcal{D}_{R_E})$). Let $\mathcal{X}'$ denote the set of terminals that remain reachable from $s_0$ in the pruned graph $G' = (V, \{e \in E : R_E(e) \geq \tau\})$. Then:*

*(i) (Survival.) Every $x \in \mathcal{X}$ with $\beta(x) > \tau + \varepsilon$ satisfies $x \in \mathcal{X}'$.*

*(ii) (Distortion.) Assume $\mathcal{X}' \neq \varnothing$, and let $p(x) := R(x)/Z$, $p_\tau(x) := R(x) \cdot \mathbf{1}\{x \in \mathcal{X}'\}/(Z - \Delta_\tau)$ with $\Delta_\tau := \sum_{x \notin \mathcal{X}'} R(x)$ and $\delta_\tau := \Delta_\tau/Z$. Then*

$$\|p - p_\tau\|_{\text{TV}} = \delta_\tau \leq \mathbb{P}_{x \sim p}[\beta(x) \leq \tau + \varepsilon],$$

*and $p_\tau(x)/p_\tau(y) = R(x)/R(y)$ for every $x, y \in \mathcal{X}'$.*

*Proofs appear in Appendix B (Theorems B.9 and B.11; Corollary B.10).*

For TD-GFN, high-bottleneck modes therefore survive any reasonable choice of $K$, and on the surviving terminals the original reward-proportional target is preserved exactly up to renormalization. Following this criterion, we remove all edges disconnected from the root node $s_0$, yielding a pruned subgraph $G' = (V', E')$ and an updated terminal set $\mathcal{X}'$. Notably, pruning occurs over the entire environment DAG instead of being restricted to observed transitions. This allows the model to retain paths that may lead to high-reward regions even if they are unobserved in the dataset.

This strategy focuses learning on informative regions and improves efficiency by reducing model fitting complexity (Zahavy et al., 2018; Zhang et al., 2020). The resulting subgraph defines a compact yet expressive action space that prioritizes high-reward trajectories. As illustrated in Figure 1(c), this approach facilitates generalization beyond offline data while remaining anchored in reliable signals.

To improve training efficiency on the pruned graph, we propose a prioritized backward sampling mechanism that constructs trajectories in reverse from terminal nodes. Terminal states are sampled from the intersection of $\mathcal{X}'$ and the set of dataset objects $\{s_{T_i}\}_{i=1}^M$ with probability proportional to their ground-truth rewards. From each sampled terminal node $x$, we recursively perform backward sampling toward the root $s_0$ using a learned backward policy $\mathcal{P}_B$, defined as:

$$\mathcal{P}_B(s_t|s_{t+1}) = \frac{\exp\{R_E(s_t, s_{t+1})\}}{\sum_{(s, s_{t+1}) \in E'} \exp\{R_E(s, s_{t+1})\}}, \quad (7)$$

This formulation aligns with the reward shaping principles in Equation (1). By directing policy updates toward regions prioritized by both terminal and edge rewards, this strategy reinforces the GFlowNet inductive bias of allocating sampling effort in proportion to reward, thereby enhancing sample efficiency. Moreover, combined with the pruning method

above, this approach helps mitigate the under-exploitation of terminal nodes discussed in Jang et al. (2024).

Finally, the policy is optimized on the pruned DAG using the sampled trajectories and their terminal rewards from the offline dataset. Crucially, the gradient-based updates rely exclusively on these recorded ground-truth values. This approach sidesteps the fitting errors inherent in proxy reward models and insulates the gradients from potential inaccuracies in the edge rewards learned during the IRL phase.

By leveraging reward-guided pruning to steer the policy away from low-utility regions and employing prioritized backward sampling to strategically allocate attention, we effectively achieve exploration of unobserved high-utility regions during evaluation. This paradigm is fundamentally distinct from the trial-and-error exploration inherent in proxy-based methods. Notably, the contributions of TD-GFN are orthogonal to the specific training strategies employed after trajectory sampling, as discussed in Appendix E.8.

## 4. Experiments

In this section, we empirically demonstrate the effectiveness of the proposed TD-GFN through extensive experiments. Specifically, we aim to answer the following questions: (i) Can TD-GFN effectively sample from the target reward distribution, and generate high-reward and diverse samples? (ii) Is TD-GFN training efficient? (iii) Does TD-GFN consistently improve performance across different scenarios? We also conduct ablation studies to evaluate the individual contributions of each component within our method, as presented in Appendix E.7. Comprehensive task descriptions and implementation details of the algorithm are provided in Appendix D. Further experimental results on additional real-world datasets can be found in Appendix E.1.

### 4.1. Hypergrid

We begin with experiments on the Hypergrid task (Bengio et al., 2021), a $D$-dimensional grid environment with side length $H$, resulting in a discrete space of size $H^D$. High-reward modes are narrowly concentrated near the $2^D$ corners, making the task particularly challenging for both exploration-based and offline learning algorithms. Our main experiments are conducted on a $8^4$ Hypergrid ($H = 8$, $D = 4$) following the setup in Bengio et al. (2021). We also provided results on larger Hypergrids in Appendix E.3. Additionally, in Appendix E.4, we visualize a $8 \times 8$ Hypergrid instance and illustrate the TD-GFN workflow on this task.

Due to the nascent nature of proxy-free GFlowNet training, we evaluate TD-GFN against a diverse set of baselines. These include the primary existing proxy-free method, COFlowNet (Zhang et al., 2025b), as well as popular offline RL algorithms such as CQL (Kumar et al., 2020) and

IQL (Kostrikov et al., 2022). We also compare against imitation learning methods, specifically Behavior Cloning (BC) and GAIL (Ho & Ermon, 2016). Additionally, we include *Dataset-GFN*, which serves as a vanilla proxy-free variant by training a GFlowNet directly on offline trajectories without modification. To ensure a fair comparison with COFlowNet, which relies on the flow matching (FM) objective (Bengio et al., 2021), we adopt the FM objective (Equation (8)) for our experiments. Because TD-GFN's pruning and prioritized backward sampling operate before the objective computation, they are architecturally orthogonal to the GFlowNet objective: Appendix E.8 reports comparable gains when TD-GFN is paired with TB, SubTB, and DB, and Appendix E.11 contrasts these against the same objectives applied naively to offline data. Furthermore, we train GAIL using the rebalanced dataset described in Section 3.1. This rebalanced approach improves performance relative to the version trained on the original dataset, and we evaluate it using the imitation policy from Algorithm 2.

Each algorithm is evaluated using two standard metrics (Bengio et al., 2021; Pan et al., 2023b; Zhang et al., 2024; 2025b). The first is *Empirical L1 Error*, which measures the $\mathcal{L}_1$ distance between the empirical distribution $\pi(x)$ from model samples and the normalized reward distribution $p(x) = R(x)/Z$, computed as $\mathbb{E}[|\pi(x) - p(x)|]$. The second is *Modes Found*, defined as the number of distinct reward modes discovered during training.

We construct an *Expert* dataset comprising $1,500$ trajectories using a well-trained GFlowNet policy to simulate data typically collected by domain experts. We adopt GFlowNet-generated datasets to provide a controlled and reproducible environment for simulating behavior policies with different optimality levels. This strategy is standard within the offline reinforcement learning community (Fu et al., 2020; Gulcehre et al., 2020). As demonstrated in Section 4.2, TD-GFN also delivers compelling results on real-world data. We provide a more detailed discussion of this approach and an analysis of the value of trajectory structure in Appendix F.

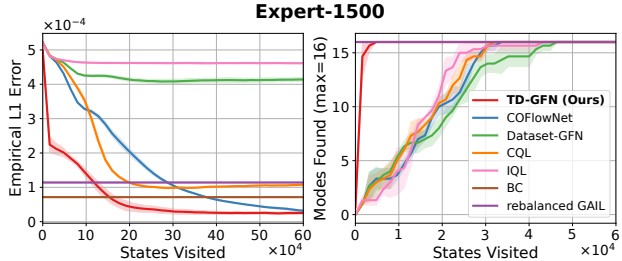

*Figure 3.* Performance comparison on the $8^4$ Hypergrid task. Results are obtained using $1,500$ trajectories from the *Expert* dataset for policy training. The solid line and shaded region represent the mean and standard deviation, respectively. This convention is maintained for all subsequent figures.

As illustrated in Figure 3, TD-GFN exhibits superior sample efficiency. It identifies all 16 modes with fewer than $5,000$ state visits, which represents a **six-fold** improvement over baselines. Furthermore, it converges to the lowest *Empirical L1 Error* over twice as fast as the most competitive baseline, COFlowNet. We include a comprehensive analysis of the other baselines in Appendix E.9.

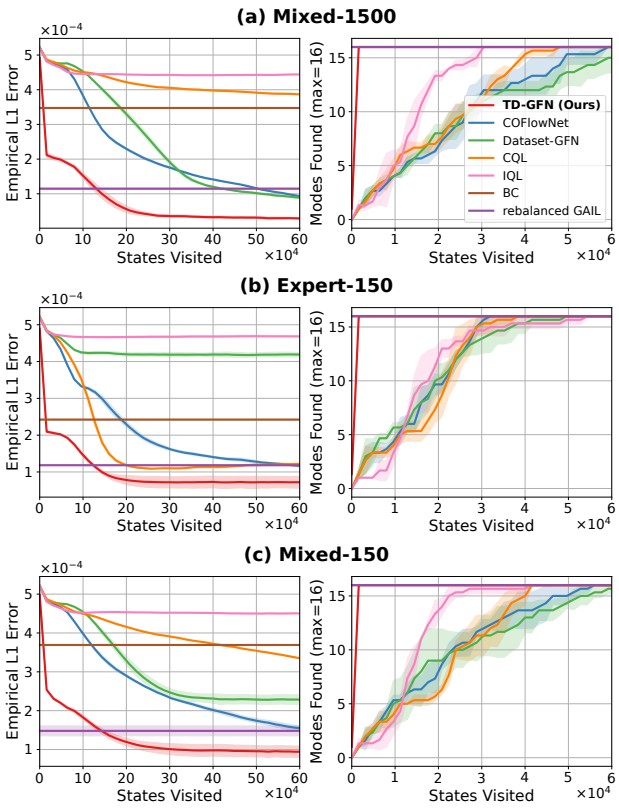

*Figure 4.* Performance comparison on the $8^4$ Hypergrid task. Labels in the format *Dataset-N* indicate training conducted with $N$ trajectories from the specified dataset.

**Robustness to Dataset Uncertainty.** We first assess the robustness of TD-GFN under dataset uncertainty by introducing greater variability in the training data. In the *Mixed* setting, we augment the *Expert* dataset with an equal number of trajectories generated by a random policy similar to Hong et al. (2023); Cao et al. (2024), thereby introducing noise. In another setting, we significantly reduce the number of available trajectories—using only one-tenth of the original dataset—to simulate data scarcity as an additional source of uncertainty (Depeweg et al., 2018).

As illustrated in Figure 4, TD-GFN demonstrates strong resilience under both conditions. It consistently models the target distribution more accurately and identifies high-reward modes significantly faster than baseline methods. These results emphasize TD-GFN's ability to maintain performance in the presence of noisy and limited data. Appendix E.2

provide additional evaluations under extreme data scarcity.

**Robustness to Behavior Policy Quality.** We also examine the sensitivity of TD-GFN to the quality of the behavior policy used during data collection. To this end, we construct two offline datasets. The first, denoted *Median*, is collected using a suboptimal GFlowNet trained for only half the number of steps required for convergence. The second, denoted *Bad*, is generated by a GFlowNet trained with an inverted reward function $IR = R^{-0.1}$, while the dataset still contains the true rewards.

As shown in Figure 5, TD-GFN adapts effectively in both scenarios, achieving faster alignment with the target reward distribution and recovering all reward modes earlier than baseline methods. Notably, when the behavior policy diverges substantially from the expert (*Bad*), TD-GFN exhibits significantly more accurate distribution modeling, underscoring its robustness under adverse conditions.

Notably, the 150 trajectories from the *Mixed* dataset and $1,500$ from the *Bad* dataset cover only 12 and 6 out of the 16 reward modes, respectively. This confirms that our experiments evaluate the model's ability to generate novel states unobserved in the dataset, rather than merely assessing its tendency to overfit the offline data.

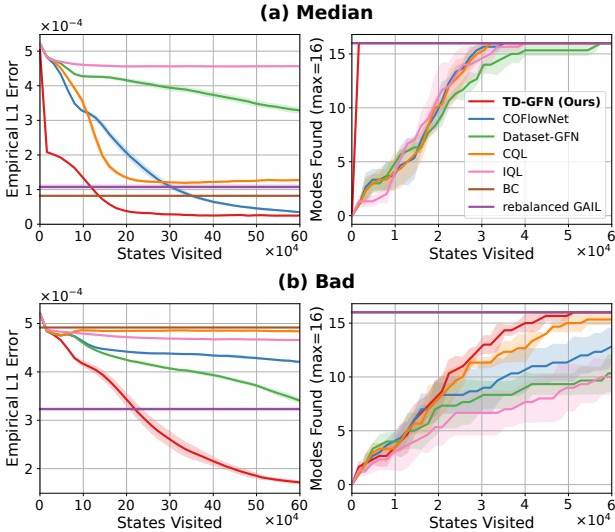

*Figure 5.* Performance comparison on the $8^4$ Hypergrid task. Policies are trained using the *Median* and *Bad* datasets, each consisting of $1,500$ trajectories.

### 4.2. Biosequence Design

We next evaluate TD-GFN on a more challenging task involving the design of anti-microbial peptides (AMPs, i.e., short protein sequences) introduced in Jain et al. (2022). The experimental setup utilizes two datasets curated from the DBAASP database (Pirtskhalava et al., 2021). One dataset

is used for training a reward proxy, while the other serves as an oracle to provide ground-truth labels.

Using the proxy reward model provided in Jain et al. (2022), we train a Proxy-GFN and compare its performance with TD-GFN, as well as with other baseline methods previously evaluated on the Hypergrid task (Section 4.1). All methods are trained on the same dataset used for proxy learning, which includes 3,219 positive AMPs and 4,761 non-AMPs.

We generate 5,000 sequences using each learned policy and report the reward and diversity metrics for the top 100 sequences ranked by reward, as shown in Figure 6. TD-GFN consistently produces sequences with the highest rewards and greatest diversity. Notably, although rebalanced GAIL merely imitates the policy implied by the rebalanced dataset, it still achieves strong performance—outperforming Dataset-GFN, which is trained directly on the dataset trajectories, in both reward and diversity. These findings highlight the effectiveness of our dataset rebalancing strategy.

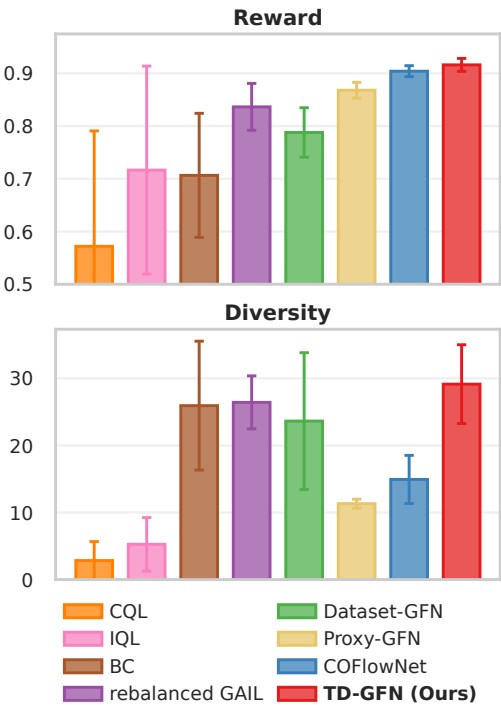

*Figure 6.* Comparison of reward and diversity among the top 100 sequences ranked by reward. The *Diversity* metric is defined in Appendix D.2. All results are reported over three random seeds.

### 4.3. Molecule Design

We further evaluate our method on the real-world molecular design benchmark introduced by Bengio et al. (2021). The training dataset simulates historical expert data, comprising 1,500 trajectories generated by a moderately trained GFlowNet policy (Behavior-GFN). These trajectories are

evaluated using the oracle provided in the original study, which was pre-trained on 300,000 molecules.

Following the architecture and methodology commonly adopted in prior GFlowNet research (Bengio et al., 2021; Malkin et al., 2022; Jang et al., 2023; Tiapkin et al., 2024), we train a proxy model on the offline dataset to optimize a GFlowNet baseline termed Proxy-GFN. We also include an Oracle-GFN that utilizes the ground-truth oracle directly as the reward function. This baseline is provided strictly for reference, as it is trained through direct environment inter-action rather than the restricted offline dataset. Furthermore, we evaluate both the flow matching (FM) COFlowNet and its quantile-augmented (QM) variant proposed by Zhang et al. (2025b), which employs quantile matching (Zhang et al., 2024) to enhance candidate diversity. We also incorporate BraVE (Landers et al., 2025), a recent offline RL algorithm designed for discrete action spaces, to extend our comparison to non-GFlowNet methods.

As summarized in Table 1, we report the average top-$k$ rewards from 5,000 generated samples along with the number of trajectories utilized for loss computation and gradient updates to reach convergence. This convergence measure reflects both the learning efficiency and the informational utility of the sampled paths for model training. Due to the instability of offline RL under sparse rewards, results for those methods reflect only their best-performing random seed. In this comparison, although BraVE converges fastest in terms of trajectory count, it fails to generate a diverse set of high-reward candidates. This limitation arises from the misalignment between the standard RL paradigm and the diversity objective of the task, which leads to significantly lower rewards than GFlowNet-based or even imitation learn-ing approaches. Excluding offline RL baselines, TD-GFN achieves the best performance in discovering high-reward molecules while requiring the fewest trajectories. Addi-tional comparisons regarding training time and computa-tional resources are provided in Appendix E.6.

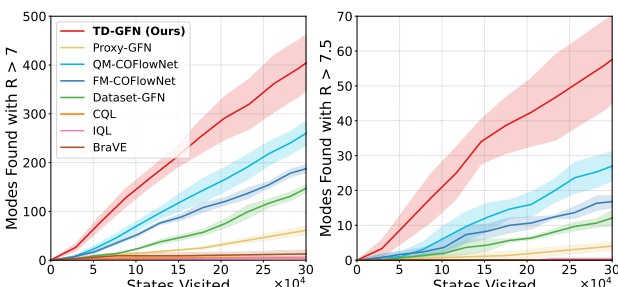

*Figure 7.* Number of high-reward modes (reward $> 7$ and $> 7.5$) discovered during training in the Molecule Design task.

Notably, the powerful intermediate guidance enables TD-GFN to match Oracle-GFN in identifying multiple high-reward molecules while decisively surpassing the behavior

*Table 1.* Performance comparison on the Molecule Design task. The *Convergence* column denotes the number of trajectories required for performance convergence. Results are reported as mean $\pm$ standard deviation over three random seeds, with bold values indicating the best performance in each column. The *Dataset* row provides statistics for the $1,500$ samples in the training dataset. Methods positioned above the thin horizontal line are included for reference and are not directly comparable to the main experimental offline baselines.

| Method | Reward-10 ($\uparrow$) | Reward-100 ($\uparrow$) | Reward-1000 ($\uparrow$) | Convergence ($\downarrow$) |
|---|---|---|---|---|
| Dataset | $7.420 \pm 0.088$ | $6.968 \pm 0.226$ | $5.757 \pm 0.635$ | / |
| Behavior-GFN | $7.534 \pm 0.066$ | $7.220 \pm 0.165$ | $6.504 \pm 0.348$ | / |
| Oracle-GFN | $7.718 \pm 0.014$ | $7.408 \pm 0.021$ | $6.801 \pm 0.023$ | $44.141 \times 10^4$ |
| CQL | $7.069$ | $6.643$ | $5.401$ | $0.803 \times 10^4$ |
| IQL | $6.902$ | $5.980$ | $4.628$ | $4.104 \times 10^4$ |
| BraVE | $7.271$ | $6.650$ | $5.590$ | $\mathbf{0.645 \times 10^4}$ |
| BC | $7.652 \pm 0.053$ | $7.223 \pm 0.035$ | $6.459 \pm 0.016$ | / |
| GAIL | $7.528 \pm 0.068$ | $7.152 \pm 0.085$ | $6.406 \pm 0.033$ | / |
| Proxy-GFN | $7.625 \pm 0.063$ | $7.281 \pm 0.067$ | $6.636 \pm 0.097$ | $43.735 \times 10^4$ |
| Dataset-GFN | $7.550 \pm 0.045$ | $7.198 \pm 0.018$ | $6.474 \pm 0.018$ | $6.030 \times 10^4$ |
| FM-COFlowNet | $7.582 \pm 0.057$ | $7.201 \pm 0.015$ | $6.485 \pm 0.016$ | $5.829 \times 10^4$ |
| QM-COFlowNet | $7.611 \pm 0.020$ | $7.296 \pm 0.022$ | $6.638 \pm 0.010$ | $4.423 \times 10^4$ |
| **TD-GFN (Ours)** | $\mathbf{7.733 \pm 0.036}$ | $\mathbf{7.450 \pm 0.037}$ | $\mathbf{6.810 \pm 0.035}$ | $\mathbf{2.749 \times 10^4}$ |

policy. This efficiency demonstrates its capacity to transcend training data limitations and suggests strong potential for iterative evolution. Furthermore, TD-GFN converges using 20 times fewer trajectories. While this results from efficient sampling and the pruned environment, it also highlights a fundamental distinction between training paradigms. Unlike online methods that require extensive trial-and-error because rewards are unknown until a trajectory is complete, offline methods can leverage recorded terminal rewards to prioritize high-utility paths during sampling.

To evaluate molecular diversity, we measure the number of distinct high-reward modes identified by each algorithm. Modes are defined as clusters of high-reward molecules determined by Tanimoto similarity (Bajusz et al., 2015). As illustrated in Figure 7, TD-GFN consistently identifies 1.5 to 2 times more high-reward modes than the strongest baselines, demonstrating superior exploration capabilities. Notably, despite using DAG pruning, TD-GFN avoids overfitting to narrow regions of the search space. Instead, it guides the policy toward a broader set of high-reward regions by balancing targeted exploitation with diverse exploration. We provide additional details on the Tanimoto similarity among sampled molecules in Appendix E.10.

## 5. Conclusion

We presented **Trajectory-Distilled GFlowNet (TD-GFN)**, a robust proxy-free framework for learning GFlowNets from offline data without requiring out-of-distribution reward queries. By utilizing edge-level guidance learned through inverse reinforcement learning, TD-GFN improves performance and training efficiency via DAG pruning and priori-

tized backward trajectory sampling. Extensive experiments across diverse benchmarks demonstrate state-of-the-art results, establishing TD-GFN as a scalable and efficient solution that effectively bridges the gap between limited offline datasets and the complex exploration requirements of generative modeling in real-world applications.

## Acknowledgments

This work was supported by the National Natural Science Foundation of China Grant 52494974.

## Impact Statement

TD-GFN targets offline GFlowNet training for compositional generation, with applications in molecule and drug discovery, biological sequence design, and other domains where active reward queries are prohibitive. By improving sample efficiency and diversity from limited offline data, our framework can reduce the number of reward queries needed in costly wet-lab pipelines. The same technical advances, however, can be misused to design harmful molecules or sequences; our work relies on mitigations standard to this line of research, including responsible release of datasets, dual-use review for downstream applications, and domain-specific safety filters at deployment. No other specific societal risks are identified.

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

# A. Algorithm Pseudocode

---

**Algorithm 1** Trajectory-Distilled GFlowNet (TD-GFN)

---

1: **Input:** Offline dataset $\mathcal{D} = \{\tau_i = (s_0, \ldots, s_{T_i}, R(s_{T_i}))\}_{i=1}^M$; number of training iterations $N$; batch size $B$; GFlowNet learning rate $\eta$.
2: **Phase 1: Edge Reward Extraction**
3: Learn edge rewards $R_E(s, s')$ and the imitation policy $\pi_\psi$ via Algorithm 2.
4: **Phase 2: DAG Pruning**
5: Collect edge rewards $\mathcal{D}_{R_E}$ using $R_E(s, s')$ and $\pi_\psi$.
6: Prune the environment DAG $G = (V, E)$ based on the thresholding rule defined in Equation (6).
7: Remove disconnected edges to obtain the pruned DAG $G' = (V', E')$.
8: **Phase 3: Policy Training**
9: Initialize the parameterized GFlowNet policy $\mathcal{P}_F$.
10: **for** $n = 1$ to $N$ **do**
11:     Initialize a batch of trajectories $\mathcal{T} = \varnothing$.
12:     **for** $j = 1$ to $B$ **do**
13:         Sample a terminal node $x_j \in \mathcal{X}' \cap \{s_{T_i}\}_{i=1}^M$ with probability $P(x) \propto R(x)$.
14:         Initialize a trajectory $\tau_j = \varnothing, s_{now} = x_j, s_{pre} = x_j$.
15:         **while** $s_{now} \neq s_0$ **do**
16:             Sample $s_{pre} \sim \mathcal{P}_B(\cdot | s_{now})$, where $\mathcal{P}_B$ is defined in Equation (7) over $G'$.
17:             $\tau_j = \tau_j \cup \{(s_{pre} \rightarrow s_{now})\}$.
18:             $s_{now} = s_{pre}$
19:         **end while**
20:         $\mathcal{T} = \mathcal{T} \cup \{\tau_j\}$.
21:     **end for**
22:     Optimize $\mathcal{P}_F$ over $G'$ with sampled trajectories $\mathcal{T}$.
23: **end for**
24: **Output:** Trained GFlowNet policy $\mathcal{P}_F$.

---

**Algorithm 2** Edge Reward Extraction

---

1: **Input:** Offline dataset $\mathcal{D} = \{\tau_i = (s_0, \ldots, s_{T_i}, R(s_{T_i}))\}_{i=1}^M$; number of training iterations $N'$; batch size $B'$; policy learning rate $\alpha$; discriminator learning rate $\beta$.
2: Initialize policy $\pi_\psi$ and discriminator $D_\phi : E \rightarrow (0, 1)$ with random parameters $\psi$ and $\phi$.
3: **for** $n = 1$ to $N'$ **do**
4:     Initialize a batch of trajectories $\mathcal{T}' = \varnothing$.
5:     **for** $j = 1$ to $B'$ **do**
6:         Sample a trajectory $\tau_j \in \mathcal{D}$ with probability $P(\tau_j) \propto R(s_{T_j})$ .
7:         $\mathcal{T}' = \mathcal{T}' \cup \{\tau_j\}$.
8:     **end for**
9:     Extract edge transitions $\{(s, s')\}$ from $\mathcal{T}'$ as a minibatch $\overline{E}$.
10:     Compute $\nabla_\phi L$ on $\overline{E}$; update $\phi \leftarrow \phi - \beta \nabla_\phi L$.
11:     Compute $\nabla_\psi L$ on $\overline{E}$; update $\psi \leftarrow \psi - \alpha \nabla_\psi L$ or apply a policy gradient method.
12: **end for**
13: Extract the edge reward function $R_E$ according to Equation (5).
14: **Output:** Edge reward function $R_E$; imitation policy $\pi_\psi$.

---

# B. Theoretical Analysis of IRL Edge Rewards and Pruning

This appendix provides a formal analysis of the two core mechanisms of TD-GFN: (i) the edge reward learned by IRL and its connection to the expert backward policy, and (ii) the robustness of the reward-guided pruning rule.

## B.1. Setup and Notation

We work in the $\gamma = \lambda = 1$ entropy-regularized RL regime of Equations (1)–(2). Let $G = (V, E)$ be a finite DAG with root $s_0$, terminal set $\mathcal{X} \subseteq V$, and an added absorbing state $s_f$. Define the extended edge set

$$\bar{E} := E \cup \{(x, s_f) : x \in \mathcal{X}\}.$$

Assume the expert distribution used in the IRL objective (Equation (3)) is generated by a fixed GFlowNet with forward policy $\mathcal{P}_F^*$, backward policy $\mathcal{P}_B^*$, and positive terminal reward $R : \mathcal{X} \to \mathbb{R}_{>0}$, so that

$$\Pr_{\mathcal{P}_F^*}(x) = \frac{R(x)}{Z}, \qquad Z := \sum_{x \in \mathcal{X}} R(x).$$

Define the *reach probability*

$$\nu^*(s) := \Pr_{\mathcal{P}_F^*}(\text{trajectory visits } s),$$

the *state flow*

$$F^*(s) := Z \cdot \nu^*(s),$$

and the *edge flow*

$$F^*(s, s') := F^*(s) \, \mathcal{P}_F^*(s'|s).$$

Then $F^*(x) = R(x)$ for terminals, and $\mathcal{P}_B^*(s|s') = F^*(s, s')/F^*(s')$.

For an edge reward $r$ on $\bar{E}$, write

$$J_r(\pi) := \mathcal{H}(\pi) + \mathbb{E}_\pi \left[ \sum_{t=0}^{\infty} r(s_t, s_{t+1}) \right].$$

The tabular population max-causal-entropy IRL problem corresponding to Equation (3) is

$$\min_r \left( \max_\pi J_r(\pi) - J_r(\mathcal{P}_F^*) \right).$$

Throughout, we adopt the standard finite-horizon GFlowNet conventions: (i) every state $s \in V$ is reachable from $s_0$ under $\mathcal{P}_F^*$ (so that $F^*(s) > 0$); (ii) every nonterminal state has at least one outgoing edge in $E$ leading toward $\mathcal{X}$, so each trajectory terminates in finitely many steps at some $x \in \mathcal{X}$, after which the deterministic transition $x \to s_f$ is appended (carrying the boundary reward $r(x, s_f) = \log R(x)$ from Equation (1)) and then $s_f$ absorbs the trajectory with no further contribution to $J_r(\pi)$; and (iii) $\mathcal{P}_F^*(s' \mid s) > 0$ for all $(s, s') \in E$ (otherwise we restrict to the support). Under these conventions, the infinite sum in $J_r(\pi)$ is in fact a finite sum: each trajectory accumulates rewards along its (finite) edges in $E$, plus the single boundary term $\log R(x)$ at the terminal transition.

We write $r^*(s, s') := \log \mathcal{P}_B^*(s|s')$ for the *canonical edge reward* and $R_E(s, s')$ for the edge reward learned by IRL (Equation (5)).

## B.2. IRL Edge Rewards and the Backward Policy

**Proposition B.1** (Trajectory Reweighting Preserves the Backward Policy). *Suppose the raw offline data are sampled from a GFlowNet $(\mathcal{P}_F, \mathcal{P}_B, R)$. Reweight trajectories by a positive terminal factor $w(x)$:*

$$q_w(\tau) \propto w(x_\tau) \cdot p(\tau),$$

*where $x_\tau$ is the terminal state of $\tau$. Then $q_w$ is exactly the path law of another GFlowNet with the **same backward policy** $\mathcal{P}_B$ and terminal weight $\widetilde{R}(x) = w(x)R(x)$.*

*In particular, if Section 3.1 reweights by $w(x) = R(x)$, then the pseudo-expert has terminal weight $R(x)^2$, but its nonterminal canonical reward is still $\log \mathcal{P}_B$.*

*Proof.* For a trajectory $\tau = (s_0, s_1, \ldots, s_T = x)$,

$$p(\tau) = \prod_{t=0}^{T-1} \mathcal{P}_F(s_{t+1}|s_t).$$

Using detailed balance $F(s_t)\mathcal{P}_F(s_{t+1}|s_t) = F(s_{t+1})\mathcal{P}_B(s_t|s_{t+1})$,

$$p(\tau) = \prod_{t=0}^{T-1} \frac{F(s_{t+1})\mathcal{P}_B(s_t|s_{t+1})}{F(s_t)} = \frac{F(x)}{F(s_0)} \prod_{t=0}^{T-1} \mathcal{P}_B(s_t|s_{t+1}).$$

Since $F(x) = R(x)$ and $F(s_0) = Z := \sum_x R(x)$,

$$p(\tau) = \frac{R(x)}{Z} \prod_{t=0}^{T-1} \mathcal{P}_B(s_t|s_{t+1}).$$

Therefore

$$q_w(\tau) = \frac{w(x)R(x)}{\sum_y w(y)R(y)} \prod_{t=0}^{T-1} \mathcal{P}_B(s_t|s_{t+1}),$$

which is exactly the trajectory law of a GFlowNet with terminal weight $\widetilde{R}(x) = w(x)R(x)$ and the same backward policy $\mathcal{P}_B$. $\qquad\square$

**Theorem B.2** (The Canonical Edge Reward). *Define the canonical reward*

$$r^*(s, s') = \begin{cases} \log \mathcal{P}_B^*(s|s'), & (s, s') \in E, \\ \log R(s), & s \in \mathcal{X}, \ s' = s_f. \end{cases}$$

*Then the entropy-regularized RL problem has soft value*

$$\Phi^*(s) = \log F^*(s), \qquad \Phi^*(s_f) = 0,$$

*and its soft-optimal policy is exactly $\mathcal{P}_F^*$.*

*Proof.* For a terminal $x$, the only outgoing edge is $x \to s_f$, so

$$e^{r^*(x, s_f) + \Phi^*(s_f)} = e^{\log R(x)} = R(x) = F^*(x),$$

hence $\Phi^*(x) = \log F^*(x)$.

For a nonterminal state $s$,

$$\sum_{s'} e^{r^*(s, s') + \Phi^*(s')} = \sum_{s'} e^{\log \mathcal{P}_B^*(s|s') + \log F^*(s')} = \sum_{s'} \mathcal{P}_B^*(s|s')F^*(s').$$

By detailed balance, $\mathcal{P}_B^*(s|s')F^*(s') = F^*(s)\mathcal{P}_F^*(s'|s)$, so

$$\sum_{s'} e^{r^*(s, s') + \Phi^*(s')} = F^*(s) \sum_{s'} \mathcal{P}_F^*(s'|s) = F^*(s).$$

Therefore $\Phi^*(s) = \log F^*(s)$.

The induced soft-optimal policy is

$$\pi^*(s'|s) = e^{r^*(s, s') + \Phi^*(s') - \Phi^*(s)} = e^{\log \mathcal{P}_B^*(s|s') + \log F^*(s') - \log F^*(s)} = \mathcal{P}_F^*(s'|s),$$

again by detailed balance. $\qquad\square$

**Corollary B.3** (Interpretation of the Edge Reward). *For every edge $(s, s') \in E$,*

$$r^*(s, s') = \log \mathcal{P}_B^*(s|s') = \log \frac{F^*(s, s')}{F^*(s')}.$$

*That is, the canonical edge reward is the log conditional incoming-flow score. For a fixed child $s'$, ranking parents by $r^*(s, s')$ is exactly ranking them by their contribution $F^*(s, s')$ to the expert flow at $s'$. Low $r^*(s, s')$ means "conditioned on reaching $s'$, the expert almost never came through $s$," which provides a principled justification for using $R_E$ as a pruning criterion.*

**Theorem B.4** (Complete Characterization of IRL Solutions). *Restrict the IRL search to rewards on $\bar{E}$ whose terminal boundary is fixed to $r(x, s_f) = \log R(x)$ for all $x \in \mathcal{X}$. Then a reward $r$ on $\bar{E}$ is a global minimizer of the population IRL problem (Equation (3)) if and only if there exists a function $\Phi : V \cup \{s_f\} \to \mathbb{R}$ with*

$$\Phi(s_f) = 0, \qquad \Phi(x) = \log R(x) \quad \forall x \in \mathcal{X},$$

*such that for every nonterminal edge $(s, s') \in E$,*

$$r(s, s') = \log \mathcal{P}_F^*(s'|s) + \Phi(s) - \Phi(s').$$

*Equivalently, every such IRL-optimal reward has the form*

$$r(s, s') = r^*(s, s') + u(s) - u(s')$$

*for some potential $u : V \cup \{s_f\} \to \mathbb{R}$ satisfying $u(x) = u(s_f) = 0$ on terminals and the absorbing state.*

*Proof.* Let $\mathcal{L}(r) := \max_\pi J_r(\pi) - J_r(\mathcal{P}_F^*)$. Since the maximization includes $\pi = \mathcal{P}_F^*$, one always has $\mathcal{L}(r) \geq 0$. Therefore $r$ is a global minimizer if and only if $\mathcal{L}(r) = 0$, i.e., $\mathcal{P}_F^*$ is itself optimal for reward $r$.

By standard soft Bellman optimality, $\mathcal{P}_F^*$ is soft-optimal for $r$ if and only if there exists a soft value $\Phi$ such that

$$\mathcal{P}_F^*(s'|s) = e^{r(s,s') + \Phi(s') - \Phi(s)}.$$

Rearranging gives $r(s, s') = \log \mathcal{P}_F^*(s'|s) + \Phi(s) - \Phi(s')$.

For a terminal $x$, the only outgoing edge is $x \to s_f$, so $r(x, s_f) = \Phi(x) - \Phi(s_f) = \Phi(x)$, and fixing the terminal reward to $\log R(x)$ implies $\Phi(x) = \log R(x)$.

Conversely, given any $\Phi$ with those boundary values, define $r$ by $r(s, s') = \log \mathcal{P}_F^*(s'|s) + \Phi(s) - \Phi(s')$. Then for every nonterminal state $s$,

$$\sum_{s'} e^{r(s,s') + \Phi(s')} = \sum_{s'} e^{\log \mathcal{P}_F^*(s'|s) + \Phi(s)} = e^{\Phi(s)},$$

so $\Phi$ satisfies the soft Bellman equation, and the induced soft-optimal policy is $\mathcal{P}_F^*(s'|s)$. Hence $\mathcal{L}(r) = 0$.

Finally, Theorem B.2 gives $r^*(s, s') = \log \mathcal{P}_F^*(s'|s) + \log F^*(s) - \log F^*(s')$. Subtracting,

$$r(s, s') - r^*(s, s') = \big(\Phi(s) - \log F^*(s)\big) - \big(\Phi(s') - \log F^*(s')\big).$$

Setting $u(s) := \Phi(s) - \log F^*(s)$, since $F^*(x) = R(x)$ and $\Phi(x) = \log R(x)$, we have $u(x) = 0$; also $u(s_f) = 0$. Thus $r = r^* + u(s) - u(s')$. $\qquad \square$

**Theorem B.5** (Unique Recovery under the Canonical Gauge). *Among all IRL-optimal rewards (in the sense of Theorem B.4, i.e., with the fixed terminal boundary $r(x, s_f) = \log R(x)$), the unique one additionally satisfying the **parent-normalization gauge***

$$\sum_{p \in \mathrm{Pa}(s')} e^{r(p, s')} = 1 \qquad \forall s' \in V \setminus \{s_0\}$$

*is exactly*

$$r^*(s, s') = \log \mathcal{P}_B^*(s|s').$$

*Note that $r^*$ is already parent-normalized:* $\sum_{p\in\mathrm{Pa}(s')} e^{r^*(p,s')} = \sum_p \mathcal{P}_B^*(p \mid s') = 1$. *Hence the parent softmax in Equation* (7) *is the identity when applied to* $r^*$, *and* $\hat{\mathcal{P}}_B = \mathcal{P}_B^*$ *exactly. For an arbitrary IRL-optimal reward* $r = r^* + u(s) - u(s')$ *with nonzero potential, the parent softmax in Equation* (7) *produces* $\hat{\mathcal{P}}_B(p \mid s') \propto \mathcal{P}_B^*(p \mid s')\, e^{u(p)}$, *which equals* $\mathcal{P}_B^*$ *only when* $u$ *is constant over* $\mathrm{Pa}(s')$ *for every* $s'$. *Thus, when the IRL estimator* $R_E$ *approximates* $r^*$ *in sup-norm, Equation* (7) *approximates* $\mathcal{P}_B^*$ *with the quantitative TV bound given in Theorem* B.7.

**Application to the rebalanced dataset.** The IRL formulation in Equation (3) solved by TD-GFN treats the rebalanced trajectory distribution $\widetilde{\mathcal{D}}$ as the expert. By Proposition B.1, this rebalanced expert is the GFlowNet with the same backward policy $\mathcal{P}_B^*$ but with terminal weight $\widetilde{R}(x) = w(x)\, R(x)$. Theorems B.4–B.5 therefore apply with $\widetilde{R}$ in place of $R$ in the terminal boundary; the canonical edge reward on $E$ is invariant, since $r^*(s, s') = \log \mathcal{P}_B^*(s \mid s')$ depends only on $\mathcal{P}_B^*$ and not on the terminal weight. Rebalancing therefore does not affect the downstream use of $R_E$ in DAG pruning and prioritized sampling, both of which depend on $r^*$ only on nonterminal edges. The original reward-proportional target $p(x) = R(x)/Z$ used in Theorem B.11 remains based on the original $R$ – the distribution that TD-GFN's policy ultimately aims to match – and is independent of the rebalanced weight used for IRL training.

*Remark* B.6 (Gauge of GAIL-derived $R_E$). The GAIL-style estimator in Equation (5) does not automatically place $R_E$ in the parent-normalization gauge. Theorem B.5 is therefore an interpretive result identifying the canonical IRL solution $r^*$, not a claim that $R_E$ equals $r^*$ exactly. The downstream stability of TD-GFN does not rely on exact gauge canonicalization: the pruning analysis in Section B.3 only assumes a sup-norm bound $\|R_E - r^*\|_\infty \le \varepsilon$, and the perturbation bound in Theorem B.7 converts this into a TV bound for the backward policy implied by Equation (7).

*Proof.* Let $r$ be IRL-optimal and satisfy the parent-normalization gauge. By Theorem B.4, there exists $\Phi$ such that

$$r(p, s') = \log \mathcal{P}_F^*(s'|p) + \Phi(p) - \Phi(s').$$

Define $G(s) := e^{\Phi(s)}$. Then for any non-root state $s' \in V \setminus \{s_0\}$,

$$1 = \sum_{p\in\mathrm{Pa}(s')} e^{r(p,s')} = e^{-\Phi(s')} \sum_{p\in\mathrm{Pa}(s')} e^{\Phi(p)} \mathcal{P}_F^*(s'|p),$$

so

$$G(s') = \sum_{p\in\mathrm{Pa}(s')} G(p)\, \mathcal{P}_F^*(s'|p). \qquad (*)$$

We claim that every positive solution of $(*)$ has the form $G(s) = c \cdot \nu^*(s)$ for some constant $c > 0$. This is proved by induction over any topological order of the DAG. At the root, $G(s_0) = c = c \cdot \nu^*(s_0)$ since $\nu^*(s_0) = 1$. If the claim holds for all parents of $s'$, then

$$G(s') = \sum_p c \cdot \nu^*(p)\, \mathcal{P}_F^*(s'|p) = c \cdot \nu^*(s'),$$

because the reach probability of $s'$ is exactly the sum over its parents of parent-reach probability times transition probability.

Now use the terminal boundary. For every terminal $x$,

$$G(x) = e^{\Phi(x)} = R(x).$$

But also $G(x) = c \cdot \nu^*(x)$, and $\nu^*(x) = R(x)/Z$. Hence $R(x) = c \cdot R(x)/Z$, which forces $c = Z$. Therefore $G(s) = Z \cdot \nu^*(s) = F^*(s)$, so $\Phi(s) = \log F^*(s)$, and substituting back,

$$r(s, s') = \log \mathcal{P}_F^*(s'|s) + \log F^*(s) - \log F^*(s') = \log \mathcal{P}_B^*(s|s') = r^*(s, s').$$

Uniqueness follows. $\qquad\square$

**Theorem B.7** (Perturbation Bound for the Recovered Backward Policy). *Let $R_E$ be a gauge-fixed estimator of the canonical reward $r^*$ such that, with probability at least $1 - \delta$,*

$$\|R_E - r^*\|_\infty \le \varepsilon_N(\delta).$$

*Define the recovered backward policy via Equation* (7):

$$\hat{\mathcal{P}}_B(s|s') = \frac{e^{R_E(s,s')}}{\sum_{p \in \mathrm{Pa}(s')} e^{R_E(p,s')}}.$$

*Then, with the same probability,*

$$\sup_{s' \in V \setminus \{s_0\}} \mathrm{TV}\Big(\hat{\mathcal{P}}_B(\cdot|s'),\ \mathcal{P}_B^*(\cdot|s')\Big) \le \frac{e^{2\varepsilon_N(\delta)} - 1}{2}.$$

*Also, for any threshold $\tau$,*

$$r^*(e) \ge \tau + \varepsilon_N(\delta) \implies R_E(e) \ge \tau, \qquad r^*(e) \le \tau - \varepsilon_N(\delta) \implies R_E(e) \le \tau.$$

*Hence if $\varepsilon_N(\delta) \to 0$ as $N \to \infty$, then $R_E \to r^* = \log \mathcal{P}_B^*$ and $\hat{\mathcal{P}}_B \to \mathcal{P}_B^*$ uniformly.*

*Proof.* Fix a child $s'$, and write $z_p := r^*(p, s')$, $\hat{z}_p := R_E(p, s')$. By assumption, $|\hat{z}_p - z_p| \le \varepsilon_N$ for all parents $p$. Let

$$\mathcal{Z} := \sum_p e^{z_p}, \qquad \hat{\mathcal{Z}} := \sum_p e^{\hat{z}_p}.$$

Then $e^{-\varepsilon_N} \mathcal{Z} \le \hat{\mathcal{Z}} \le e^{\varepsilon_N} \mathcal{Z}$.

Therefore for each parent $p$,

$$\frac{\hat{\mathcal{P}}_B(p|s')}{\mathcal{P}_B^*(p|s')} = e^{\hat{z}_p - z_p} \cdot \frac{\mathcal{Z}}{\hat{\mathcal{Z}}} \in \left[e^{-2\varepsilon_N},\ e^{2\varepsilon_N}\right].$$

So $|\hat{\mathcal{P}}_B(p|s') - \mathcal{P}_B^*(p|s')| \le (e^{2\varepsilon_N} - 1)\mathcal{P}_B^*(p|s')$. Summing over parents gives

$$\|\hat{\mathcal{P}}_B(\cdot|s') - \mathcal{P}_B^*(\cdot|s')\|_1 \le e^{2\varepsilon_N} - 1,$$

hence $\mathrm{TV}(\hat{\mathcal{P}}_B(\cdot|s'), \mathcal{P}_B^*(\cdot|s')) \le (e^{2\varepsilon_N} - 1)/2$. Taking the supremum over $s'$ proves the first claim.

The threshold statements are immediate from the sup-norm bound: $R_E(e) \ge r^*(e) - \varepsilon_N$ and $R_E(e) \le r^*(e) + \varepsilon_N$. Uniform convergence follows when $\varepsilon_N \to 0$. $\qquad\square$

### B.3. Theoretical Motivation for Pruning

For any score function $g : E \to \mathbb{R}$ and threshold $\tau$, define the pruned graph

$$G_\tau(g) = (V, E_\tau(g)), \qquad E_\tau(g) := \{e \in E : g(e) \ge \tau\},$$

and let $\mathcal{X}_\tau(g) \subseteq \mathcal{X}$ be the terminals reachable from $s_0$ in $G_\tau(g)$.

For each terminal $x \in \mathcal{X}$, let $\mathcal{P}(x) \ne \varnothing$ denote the set of all directed paths from $s_0$ to $x$ (by the reachability assumption in Section B.2 this set is nonempty for every terminal). Define the *bottleneck score*

$$\beta_g(x) := \max_{P \in \mathcal{P}(x)} \min_{e \in P} g(e).$$

We also write $\mathcal{C}(x)$ for the (finite) family of $s_0$-*to-$x$ edge cuts*, where an edge cut $C \subseteq E$ is a finite subset of edges that intersects every path $P \in \mathcal{P}(x)$.

When the score function is the canonical edge reward $r^*$, we write $\beta(x) := \beta_{r^*}(x)$ for brevity.

**Theorem B.8** (Exact Characterization of Hard Pruning). *For any score function $g$ and any threshold $\tau$,*

$$x \in \mathcal{X}_\tau(g) \iff \beta_g(x) \ge \tau.$$

*Equivalently,*

$$\beta_g(x) = \min_{C \in \mathcal{C}(x)} \max_{e \in C} g(e).$$

*Proof.* The first equivalence is immediate: $x \in \mathcal{X}_\tau(g)$ means that there exists a path $P \in \mathcal{P}(x)$ such that every edge on $P$ satisfies $g(e) \geq \tau$. This is equivalent to $\exists P \in \mathcal{P}(x)$ with $\min_{e \in P} g(e) \geq \tau$, which is exactly $\beta_g(x) \geq \tau$.

For the cut form, fix any path $P \in \mathcal{P}(x)$ and any cut $C \in \mathcal{C}(x)$. Since $P$ must cross $C$, there exists $e \in P \cap C$. Hence

$$\min_{e' \in P} g(e') \leq g(e) \leq \max_{e'' \in C} g(e'').$$

Taking the maximum over all $P$ and then the minimum over all $C$ gives $\beta_g(x) \leq \min_{C \in \mathcal{C}(x)} \max_{e \in C} g(e)$.

For the reverse inequality, define $a := \min_{C \in \mathcal{C}(x)} \max_{e \in C} g(e)$. Consider the thresholded graph $G_a(g)$. If $x$ were not reachable from $s_0$ in $G_a(g)$, let $A$ be the set of nodes reachable from $s_0$ in $G_a(g)$. Then the frontier

$$C_A := \{(u, v) \in E : u \in A, \ v \notin A\}$$

is an $s_0$-to-$x$ cut. By construction, every edge in $C_A$ has score $< a$; otherwise its head would also be reachable. Thus $\max_{e \in C_A} g(e) < a$, contradicting the definition of $a$. Therefore $x$ is reachable in $G_a(g)$, and the first part implies $\beta_g(x) \geq a$. Combining both inequalities yields the result. $\square$

**Theorem B.9** (Robust Preservation under Estimation Error). *Let $r^*$ be the oracle edge reward and $R_E$ its learned estimate. Suppose $\|R_E - r^*\|_\infty \leq \varepsilon$. Then for any threshold $\tau$,*

$$\{x \in \mathcal{X} : \beta(x) > \tau + \varepsilon\} \ \subseteq \ \mathcal{X}_\tau(R_E) \ \subseteq \ \{x \in \mathcal{X} : \beta(x) \geq \tau - \varepsilon\}.$$

*In particular, any terminal $x$ with oracle bottleneck margin $\beta(x) - \tau > \varepsilon$ is guaranteed to survive pruning.*

*Proof.* Take any $x$ with $\beta(x) > \tau + \varepsilon$. By definition, there exists a path $P \in \mathcal{P}(x)$ such that $r^*(e) > \tau + \varepsilon$ for all $e \in P$. Therefore $R_E(e) \geq r^*(e) - \varepsilon > \tau$ for all $e \in P$. Hence all edges on $P$ survive thresholding, so $x \in \mathcal{X}_\tau(R_E)$.

Conversely, if $x \in \mathcal{X}_\tau(R_E)$, then there exists a path $P \in \mathcal{P}(x)$ with $R_E(e) \geq \tau$ for all $e \in P$. Thus $r^*(e) \geq R_E(e) - \varepsilon \geq \tau - \varepsilon$ for all $e \in P$, which implies $\beta(x) \geq \tau - \varepsilon$. $\square$

**Corollary B.10** (Mode-Basin Preservation). *A mode basin is any nonempty subset of terminals; let $\mathcal{M}_1, \ldots, \mathcal{M}_m \subseteq \mathcal{X}$ be pairwise disjoint mode basins. If for every $i$ there exists some $x_i \in \mathcal{M}_i$ such that $\beta(x_i) > \tau + \varepsilon$, then*

$$\mathcal{M}_i \cap \mathcal{X}_\tau(R_E) \neq \varnothing \quad \text{for all } i = 1, \ldots, m.$$

*So pruning cannot eliminate any of these mode basins.*

*Proof.* By Theorem B.9, each witness terminal $x_i$ survives pruning, i.e., $x_i \in \mathcal{X}_\tau(R_E)$. Since $x_i \in \mathcal{M}_i$, we have $\mathcal{M}_i \cap \mathcal{X}_\tau(R_E) \neq \varnothing$. $\square$

**Theorem B.11** (Reward-Mass Loss and Target-Distribution Distortion). *Let $\mathcal{X}' := \mathcal{X}_\tau(R_E)$ denote the set of surviving terminals, and assume $\mathcal{X}' \neq \varnothing$ (equivalently, $\delta_\tau < 1$). Define the lost reward mass and its normalized version:*

$$\Delta_\tau := \sum_{x \notin \mathcal{X}'} R(x), \qquad \delta_\tau := \frac{\Delta_\tau}{Z}.$$

*Then under the assumption of Theorem B.9,*

$$\Delta_\tau \leq \sum_{x:\ \beta(x) \leq \tau + \varepsilon} R(x).$$

*Define the original reward-proportional target $p(x) := R(x)/Z$, and the target on the pruned graph*

$$p_\tau(x) := \frac{R(x) \cdot \mathbf{1}\{x \in \mathcal{X}'\}}{Z - \Delta_\tau}.$$

*Then:*

(i) *$\|p - p_\tau\|_{\mathrm{TV}} = \delta_\tau \leq \mathbb{P}_{x \sim p}[\beta(x) \leq \tau + \varepsilon]$, where the first equality is exact and the inequality comes from the reward-mass bound above.*

*(ii) For any surviving terminals $x, y \in \mathcal{X}'$,*

$$\frac{p_\tau(x)}{p_\tau(y)} = \frac{p(x)}{p(y)} = \frac{R(x)}{R(y)}.$$

*That is, among surviving terminals, relative probabilities are unchanged; pruning only renormalizes. (When $\mathcal{X}' = \varnothing$ the conditional target $p_\tau$ is undefined; the reward-mass bound $\Delta_\tau \leq \sum_{\beta(x) \leq \tau + \varepsilon} R(x)$ still holds.)*

*Proof.* By Theorem B.9, if $x \notin \mathcal{X}'$, then $\beta(x) \leq \tau + \varepsilon$. Therefore

$$\Delta_\tau = \sum_{x \notin \mathcal{X}'} R(x) \leq \sum_{x:\; \beta(x) \leq \tau + \varepsilon} R(x).$$

Let $Z' := Z - \Delta_\tau$. For $x \in \mathcal{X}'$,

$$p_\tau(x) = \frac{R(x)}{Z'} = \frac{Z}{Z'} \cdot p(x) = \frac{1}{1 - \delta_\tau} \cdot p(x),$$

and for $x \notin \mathcal{X}'$, $p_\tau(x) = 0$. Hence for any $x, y \in \mathcal{X}'$,

$$\frac{p_\tau(x)}{p_\tau(y)} = \frac{R(x)/Z'}{R(y)/Z'} = \frac{R(x)}{R(y)}.$$

For total variation, split the sum over $\mathcal{X}'$ and $\mathcal{X} \setminus \mathcal{X}'$:

$$\sum_{x \in \mathcal{X}'} |p(x) - p_\tau(x)| = \sum_{x \in \mathcal{X}'} p(x) \left| 1 - \frac{1}{1 - \delta_\tau} \right| = (1 - \delta_\tau) \cdot \frac{\delta_\tau}{1 - \delta_\tau} = \delta_\tau,$$

$$\sum_{x \notin \mathcal{X}'} |p(x) - p_\tau(x)| = \sum_{x \notin \mathcal{X}'} p(x) = \delta_\tau.$$

Therefore $\|p - p_\tau\|_{\mathrm{TV}} = \frac{1}{2}(\delta_\tau + \delta_\tau) = \delta_\tau$.

The probability bound is the normalized version of the reward-mass bound:

$$\delta_\tau = \frac{\Delta_\tau}{Z} \leq \frac{1}{Z} \sum_{x:\; \beta(x) \leq \tau + \varepsilon} R(x) = \mathbb{P}_{x \sim p}[\beta(x) \leq \tau + \varepsilon]. \qquad \square$$

**Corollary B.12** (Count Bound on Dropped Modes)**.** *Let $\mathcal{M}_1, \ldots, \mathcal{M}_m \subseteq \mathcal{X}$ be pairwise disjoint mode basins (subsets of terminals), and let $W_{\min} := \min_i \sum_{x \in \mathcal{M}_i} R(x) > 0$ be the smallest basin reward mass. Then the number of basins completely removed by pruning is at most*

$$\frac{\Delta_\tau}{W_{\min}} \leq \frac{1}{W_{\min}} \sum_{x:\; \beta(x) \leq \tau + \varepsilon} R(x).$$

*In particular, if the right-hand side is $< 1$, no full mode basin can disappear.*

*Proof.* If a basin $\mathcal{M}_i$ is completely removed, all its reward mass contributes to $\Delta_\tau$. Since the basins are disjoint, the total mass of removed basins is at least $W_{\min}$ times the number of removed basins. $\qquad \square$

**Corollary B.13** (High-Reward Modes Survive under a Monotone Oracle Score)**.** *Suppose the oracle score $r^*$ satisfies the downstream monotonicity condition: there exists a nondecreasing function $\psi$ such that for every terminal $x$ and every edge $e$ on any path from $s_0$ to $x$,*

$$r^*(e) \geq \psi(R(x)).$$

*Then $\beta(x) \geq \psi(R(x))$ for all $x \in \mathcal{X}$. Hence if $R(x) \geq R_{\min}$ and $\psi(R_{\min}) > \tau + \varepsilon$, then $x \in \mathcal{X}_\tau(R_E)$. Therefore all terminals in the high-reward set $\mathcal{X}_{\mathrm{high}} := \{x \in \mathcal{X} : R(x) \geq R_{\min}\}$ survive pruning.*

*Proof.* For any path $P \in \mathcal{P}(x)$, the monotonicity assumption gives $r^*(e) \geq \psi(R(x))$ for every $e \in P$. Therefore $\min_{e \in P} r^*(e) \geq \psi(R(x))$. Taking the maximum over $P \in \mathcal{P}(x)$ yields $\beta(x) \geq \psi(R(x))$. The survival claim then follows from Theorem B.9. $\qquad \square$

# C. Related work

**Generative Flow Networks (GFlowNets).**   GFlowNets (Bengio et al., 2021; 2023) are a class of generative models designed to sample compositional objects from an unnormalized reward distribution via sequential action selection. They have been theoretically linked to entropy-regularized reinforcement learning (Tiapkin et al., 2024; Mohammadpour et al., 2024) and variational inference (Malkin et al., 2023; Zimmermann et al., 2023), offering a principled framework for distribution matching.

Recent research has focused on improving GFlowNet training by proposing new balance conditions (Malkin et al., 2022; Madan et al., 2023), enhancing credit assignment through intermediate rewards (Pan et al., 2023b;a; Jang et al., 2023), designing backward policies for improved sampling (Jang et al., 2024; Mohammadpour et al., 2024; Shen et al., 2023; Gritsaev et al., 2024), refining sampling and resampling strategies (Atanackovic & Bengio, 2024; Kim et al., 2024c;b; Lau et al., 2024; Madan et al., 2025), and exploring alternative training objectives (Silva et al., 2024; Hu et al., 2024b), including energy-based training approaches for diffusion-structured inference (Campbell et al., 2024). Extensions to the GFlowNet framework have expanded its applicability to stochastic dynamics (Pan et al., 2023c), continuous action spaces (Lahlou et al., 2023), non-acyclic transitions (Brunswic et al., 2024; Morozov et al., 2025), and implicit reward feedback (Chen & Mauch, 2024). These advances have broadened the scope of GFlowNets, enabling impactful applications in molecule design (Jain et al., 2023a;b; Zhu et al., 2023; Roy et al., 2023), biological sequence generation (Jain et al., 2022; Ghari et al., 2023), combinatorial optimization (Zhang et al., 2025a; 2023; Kim et al., 2024a), causal inference (Zhang et al., 2022; Deleu et al., 2022), recommendation systems (Liu et al., 2023), and fine-tuning of large language models (Hu et al., 2024a; Yu et al., 2024) and diffusion models (Liu et al., 2025; Venkatraman et al., 2024).

Our work is most closely related to recent efforts in offline and proxy-free GFlowNet training. An early approach, RO-GFlowNets (Wang et al., 2023), uses offline action probabilities to constrain the learned policy, while COFlowNet (Zhang et al., 2025b) penalizes flow on unseen edges to limit policy coverage to dataset-adjacent regions. Both methods aim to improve policy reliability by imposing coarse constraints that merely align policy behavior or coverage more closely with the dataset. However, such constraints impose blunt limitations that hinder generalization and make performance highly sensitive to data quality, as they fail to fully exploit the trajectory-level information contained in the dataset. Other related works include Atanackovic & Bengio (2024), which explores an "offline" setting but primarily focuses on evaluating sampling strategies, and the KL-weakFM loss (Brunswic et al., 2025), which enables proxy-free imitation learning. Concurrently, some research has investigated improving offline training with explicit reward signals and novel exploration strategies (Sendera et al., 2024), providing complementary perspectives to our pruning-based framework.

Conceptually, our work shares its core insight with Silva et al. (2025), which introduces Weighted Detailed Balance (WDB) to emphasize transitions with a larger impact on correctness, particularly in online GFlowNets with GNN-based parameterizations. Their analysis demonstrates that imbalances across edges can systematically affect sampling accuracy. Our work shares this motivation but takes a different route: instead of reweighting balance conditions, we employ inverse reinforcement learning (IRL) to estimate edge rewards. These rewards are then used to guide pruning and prioritized sampling in the offline setting. This distinction in methodology makes the two approaches complementary—WDB could potentially enhance offline training, while our pruning and sampling strategies may in turn benefit online methods.

**Inverse Reinforcement Learning (IRL).**   Inverse reinforcement learning (IRL) aims to recover a reward function that explains observed expert behavior in a Markov decision process (Ng et al., 2000). Among its variants, maximum causal entropy IRL is particularly noteworthy, as it addresses reward ambiguity by formulating the objective as finding a policy that maximizes entropy while matching the observed expert behavior (Ziebart et al., 2008; 2010). Building on this framework, Generative Adversarial Imitation Learning (GAIL) (Ho & Ermon, 2016) reinterprets imitation as a distribution matching problem via adversarial training, where a discriminator distinguishes expert from policy-generated trajectories, and its output serves as a learned reward for policy training. GAIL eliminates the need to explicitly solve the inner IRL loop, significantly improving scalability, and has inspired a broad class of Adversarial Imitation Learning (AIL) methods (Fu et al., 2018; Kostrikov et al., 2019; 2020; Garg et al., 2021). This line of work has since been extended in various directions, including efforts to improve reward generalization and robustness (Xu et al., 2020; Luo et al., 2022; 2024).

# D. Experimental details

## D.1. Hypergrid

The Hypergrid task is a synthetic benchmark introduced in Bengio et al. (2021) to evaluate the exploration and generalization capabilities of GFlowNet algorithms. It presents a significant challenge due to its high-dimensional, sparse reward landscape.

The environment consists of a $D$-dimensional hypercubic grid with side length $H$, defining a discrete state space of size $H^D$. Each state corresponds to a coordinate $x = (x_1, \ldots, x_D) \in \{0, \ldots, H-1\}^D$. The agent starts at the origin $x = (0, \ldots, 0)$ and can increment any individual coordinate by one at each step. A trajectory terminates when a special stop action is taken, and the reward is assigned based on the final state.

The reward function is designed to produce multiple sharp reward modes near the corners of the hypercube. Formally, the reward at a terminal state $x$ is defined as:

$$R(x) = R_0 + R_1 \prod_{d=1}^{D} \mathbb{I}\left(\left|\frac{x_d}{H} - 0.5\right| > 0.25\right) + R_2 \prod_{d=1}^{D} \mathbb{I}\left(0.3 < \left|\frac{x_d}{H} - 0.5\right| < 0.4\right),$$

where $R_0 < R_1 \ll R_2$, and $\mathbb{I}(\cdot)$ denotes the indicator function. The first term, weighted by $R_1$, introduces $2^D$ high-reward modes near the grid's corners, while the second term, weighted by $R_2$, creates even sparser regions of extremely high reward, thereby increasing the difficulty of discovering them through exploration alone.

Following the most challenging configuration proposed in Bengio et al. (2021), we set the environment parameters to $R_0 = 10^{-3}, R_1 = 0.5, R_2 = 2$. This configuration results in a highly multi-modal and sparsely rewarded environment, making it well-suited for evaluating the effectiveness of offline training strategies.

## D.2. Biosequence Design

The Anti-Microbial Peptide (AMP) Design task, introduced in Jain et al. (2022), is a realistic sequence generation benchmark aimed at designing short peptide sequences with strong anti-microbial properties.

The agent constructs sequences by selecting amino acids from a fixed vocabulary of 20 standard residues, with a maximum sequence length of 50. At each time step, the agent chooses either an amino acid or a special end-of-sequence token, proceeding in a left-to-right manner. As a result, the transition structure forms a tree, and the trajectories in offline datasets can be directly inferred by the terminal states.

Notably, when the environment DAG degenerates into a tree, the prioritized backward sampling strategy in TD-GFN simplifies to sampling trajectories from the dataset in proportion to their terminal rewards. Despite this simplification, TD-GFN remains effective on this task, which presents substantial challenges due to the large combinatorial space of valid sequences ($\approx 20^{50}$) and the extreme sparsity of high-reward regions.

In our experiments, we adopt the *Diversity* metric to evaluate algorithmic performance, following the methodology in Jain et al. (2022). For a set of biological sequences $C$, the metric is defined as:

$$\text{Diversity}(C) = \frac{\sum_{x_i, x_j \in C} \text{Lev}(x_i, x_j)}{|C|(|C| - 1)},$$

where $\text{Lev}(\cdot, \cdot)$ denotes the Levenshtein distance. All environmental parameters are kept consistent with those used in Jain et al. (2022).

## D.3. Molecule Design

We evaluate our method on a fragment-based molecular generation task targeting protein binding affinity, specifically for soluble epoxide hydrolase (sEH). The goal is to construct candidate molecules with high predicted affinity for the target protein. Molecules are generated sequentially using the junction tree framework introduced in Jin et al. (2018), where each action involves selecting an attachment atom and adding a fragment from a predefined vocabulary of 105 building blocks.

The reward is defined as the negative binding energy between the generated molecule and the target protein, as predicted by an oracle model pre-trained on $300k$ molecules, following the setup in Bengio et al. (2021). To balance reward maximization with output diversity, the final reward is modulated using a reward exponent $\omega$, such that $R'(x) = R(x)^\omega$, where $R(x)$

denotes the raw oracle prediction. All environmental parameters, including $\omega$, are kept consistent with those used in Bengio et al. (2021).

### D.4. Policy Model Architectures and Hyperparameters

Across the experimental tasks presented in this work, we adopt the model architectures listed in Table 2 as the backbone structures for the policy networks.

*Table 2.* Policy network architectures used across different experimental tasks.

| Task | Model architecture |
|------|--------------------|
| Hypergrid | MLP([input, 256, 256, output]) |
| Biosequence Design | MLP([input, 128, 128, output]) |
| Molecule Design | MPNN (Gilmer et al., 2017) v4 |

*Table 3.* Hyperparameters used across different tasks and methods.

| Method | Hyperparameter | Task | Value |
|--------|----------------|------|-------|
| All | Policy learning rate $\eta$ | Hypergrid | $1 \times 10^{-5}$ |
| | | Molecule Design | $1 \times 10^{-4}$ |
| | | Biosequence Design | $1 \times 10^{-3}$ |
| | Entropy coef. $\lambda$ | All | 0.01 |
| | Seeds | All | 0,1,2 |
| TD-GFN | Pruning coef. $K$ | Hypergrid | 7.0 |
| | | Molecule Design | 1.0 |
| | | Biosequence Design | 2.0 |
| | Actor learning rate $\alpha$ | Hypergrid | $1 \times 10^{-5}$ |
| | | Molecule Design | $1 \times 10^{-4}$ |
| | | Biosequence Design | $1 \times 10^{-3}$ |
| | Discriminator learning rate $\beta$ | Hypergrid | $3 \times 10^{-5}$ |
| | | Molecule Design | $3 \times 10^{-4}$ |
| | | Biosequence Design | $3 \times 10^{-3}$ |
| COFlowNet | Regularizing coef. 1 | Hypergrid | 1.0 |
| | | Molecule Design | 1.0 |
| | | Biosequence Design | / |
| | Regularizing coef. 2 | Hypergrid | 1.0 |
| | | Molecule Design | 10.0 |
| | | Biosequence Design | 25.0 |
| IQL | Expectile | Hypergrid | 0.9 |
| | | Molecule Design | 0.9 |
| | | Biosequence Design | 0.8 |
| BC | Loss | All | Cross Entropy Loss |

Key algorithmic hyperparameters used across different tasks and methods are summarized in Table 3. These values were selected through empirical tuning to ensure stable training and fair comparisons among baselines.[1] Specifically, the pruning threshold coefficient $K$ is tuned by selecting the value that yields the highest ratio of retained recorded transitions to retained randomly selected transitions. For the sensitivity analysis with respect to parameter $K$, please refer to Appendix E.5.

---

[1]Our implementation of COFlowNet is based on the official open-source repository available at https://github.com/yuxuan9982/COflownet.

# E. Additional experimental results

## E.1. Additional Experiments on Real-world Recommendation Datasets

To further evaluate TD-GFN on a **real-world combinatorial problem modeled as a general DAG**, we introduce an evaluation on the listwise recommendation task following Liu et al. (2023).

**Environment**  The environment is built upon the MovieLens 1M (ML1M) dataset[2], a standard and widely-used public benchmark for recommendation systems comprising real user-item interactions. The agent is tasked with autoregressively generating a list (or "slate") of $K = 6$ unique items for a user. With a total item pool of $3,706$, this presents a large-scale combinatorial generation challenge. Following Liu et al. (2023), the reward for a generated list $O$ is defined as the average of its item-wise rewards, $\mathcal{R}(u, O) = \frac{1}{K} \sum_{i \in O} \mathcal{R}(u, i)$, where item-wise rewards are mapped from user ratings (clicks, likes, stars) to a scale of $[0, 3]$.

Crucially, since the utility of a recommendation slate is invariant to the permutation of its items, any specific set of unique items can be generated via $K! = 720$ distinct sequential paths. This characteristic explicitly structures the environment as a non-degenerate DAG, distinguishing it from tree-structured tasks. For our offline setting, we filtered the public dataset to retain users with at least 25 interactions. We then constructed the training dataset by directly sampling trajectories of length $K = 6$ from these users' real-world interaction logs.

**Metrics**  We evaluate all methods using three standard metrics from the Liu et al. (2023) benchmark:

- **Average Reward**: The mean listwise reward over a large batch of generated samples.

- **Max Reward**: The reward of the single best list generated.

- **Coverage**: The total number of unique items that appear across all generated lists, measuring sample diversity.

**Results**  The performance of all methods on the offline ML1M dataset is presented in Table 4. TD-GFN consistently outperforms other offline GFN baselines; notably, it achieves reward performance competitive with the Oracle-GFN while exhibiting significantly higher and more stable diversity.

*Table 4.* Performance comparison on the ML1M listwise recommendation task. Results are reported as mean ± standard deviation over three random seeds.

| Method | Avg. Reward (↑) | Max Reward (↑) | Coverage (↑) |
|---|---|---|---|
| Oracle-GFN | **2.092 ± 0.033** | **2.930 ± 0.017** | 113.7 ± 72.0 |
| Dataset-GFN | 1.681 ± 0.202 | 2.674 ± 0.184 | 7.9 ± 0.9 |
| QM-COFlowNet | 1.787 ± 0.022 | 2.843 ± 0.020 | 119.2 ± 6.1 |
| **TD-GFN (Ours)** | **1.988 ± 0.006** | **2.914 ± 0.013** | **186.8 ± 6.3** |

## E.2. Algorithm trained on extremely limited data

To further assess the robustness of TD-GFN under extreme data scarcity, we conduct experiments on the $H = 8$, $D = 4$ Hypergrid environment using only **30** training trajectories from the *Expert/Mixed* datasets which cover merely **5** and **3** modes respectively, as illustrated in Figure 8. In this setting, algorithms are prone to overfitting due to the limited data. For example, both COFlowNet and CQL exhibit non-monotonic trends in *Empirical L1 Error*, initially decreasing but later increasing during training, indicating overfitting to the limited known regions.

In contrast, TD-GFN consistently converges toward the target distribution throughout training. Owing to the strong generalization capabilities of the learned edge rewards, it avoids overfitting to the dataset and quickly identifies high-reward modes. These results highlight TD-GFN's ability to maintain a favorable balance between exploration and exploitation, even under severely constrained data conditions.

---

[2]https://grouplens.org/datasets/movielens/1m/

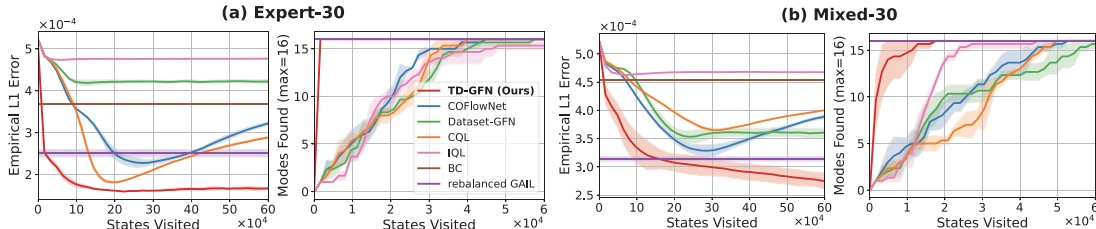

*Figure 8.* Performance comparison on the $8^4$ Hypergrid task using 30 trajectories from the *Expert* and *Mixed* datasets.

### E.3. Algorithm trained on a larger Hypergrid

We evaluate the performance of TD-GFN in more challenging settings with sparser rewards, specifically using larger $20^4$ and $256^2$ Hypergrid environments. For the $20^4$ task, as shown in Figure 9, TD-GFN achieves strong performance across all four datasets, consistently and significantly outperforming COFlowNet. Furthermore, Figure 10 demonstrates that in the $256^2$ environment, TD-GFN successfully identifies all 4 widely separated modes using only 100 training trajectories. Notably, even when trained on the "Bad" dataset which contains only 3 modes, our method effectively generalizes to discover the missing mode.

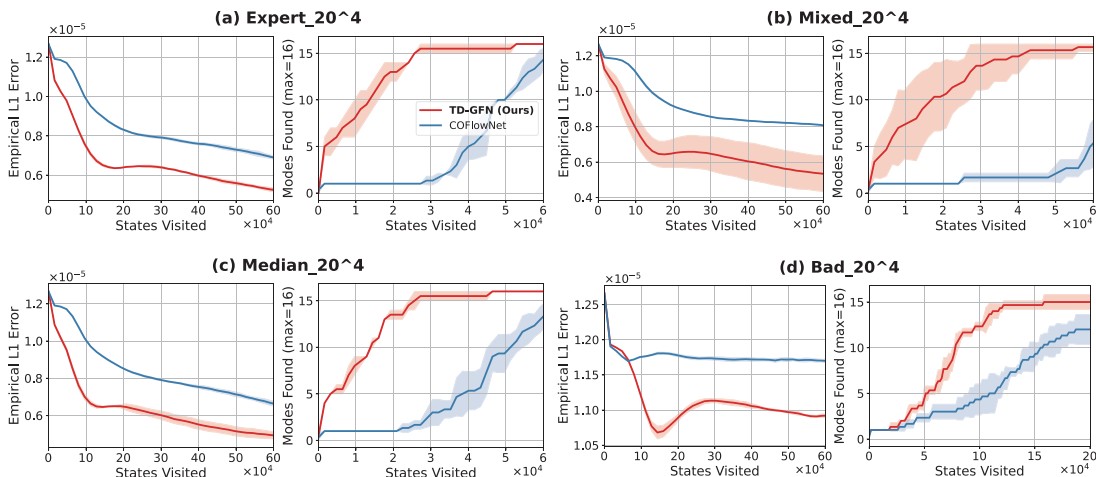

*Figure 9.* Performance comparison on the $20^4$ Hypergrid task using $1,500$ trajectories from four types of datasets.

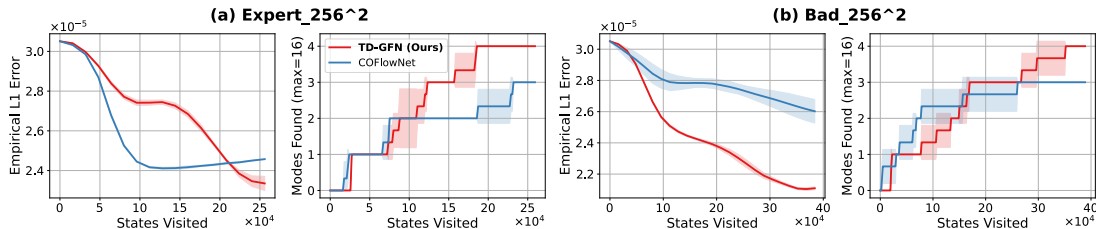

*Figure 10.* Performance comparison on the $256^2$ Hypergrid task using 100 trajectories from four types of datasets.

### E.4. Algorithm Visualization on Hypergrid

We visualize the behavior of out proposed TD-GFN in an $8 \times 8$ Hypergrid environment using a dataset comprising 400 transitions, constructed by combining trajectories collected from an expert policy ($50\%$) and a random policy ($50\%$). The edge rewards learned via IRL using Algorithm 2 are shown in Figure 11(b). Compared to the raw edge visitation frequencies from the dataset (Figure 11(a)), the learned edge rewards place greater emphasis on transitions that lead toward high-reward modes.

Notably, the learned edge rewards exhibit strong generalization: they assign meaningful values even to transitions that do not appear in the dataset (e.g., those in the topmost row), thereby guiding the policy toward out-of-distribution high-reward states, such as the modes in the top-right corner.

Figure 11(c) shows the pruned DAG based on the edge rewards in Figure 11(b). As illustrated, the resulting subgraph not only removes many transitions leading to low-reward regions—despite their presence in the dataset—but also constructs clear pathways toward high-reward modes, even when such paths were never encountered during data collection.

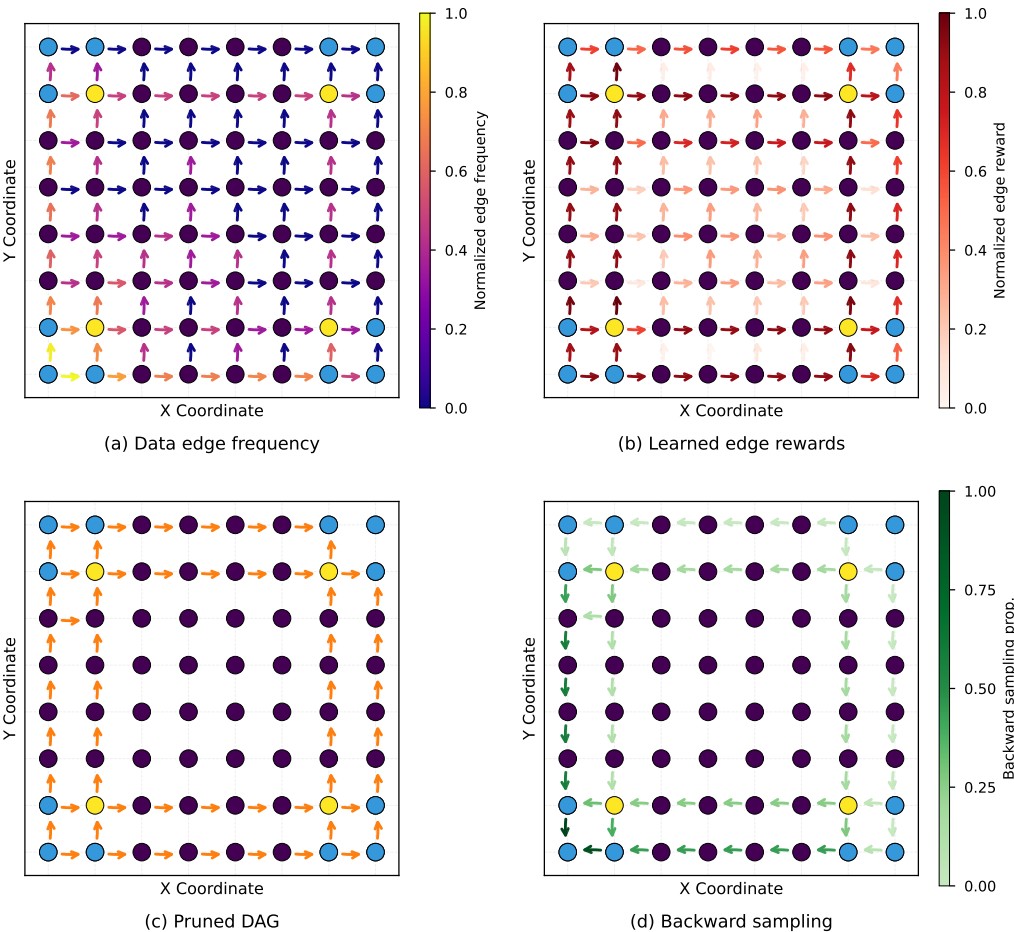

(a) Data edge frequency

(b) Learned edge rewards

(c) Pruned DAG

(d) Backward sampling

*Figure 11.* Visualization of TD-GFN in the $8 \times 8$ Hypergrid environment. (a) Normalized edge visitation frequencies in a dataset composed of 400 transitions obtained by combining trajectories collected separately from an expert policy (50%) and a random policy (50%); (b) Edge rewards learned via inverse reinforcement learning (IRL); (c) The pruned sub-DAG consisting of high-reward edges (highlighted in yellow) based on edge rewards; (d) The sampling probability of edges under backward policy induced by edge rewards and object rewards.

### E.5. Sensitivity analysis of the pruning threshold coefficient $K$

In our method, the pruning threshold hyperparameter $K$ is used to guide edge removal in a statistically principled manner (Equation (6)). It is tuned by selecting the value that yields the highest ratio of retained recorded transitions to retained randomly selected transitions in this work. Our experiments demonstrate that applying a fixed $K$ $(= 7)$ across different datasets within the same environment yields strong performance without the need for dataset-specific tuning. While future work may explore more theoretically grounded strategies for DAG pruning, here we provide a sensitivity analysis of this parameter. As shown in Figure 12, our method remains robust across different values of $K$ without catastrophic degradation.

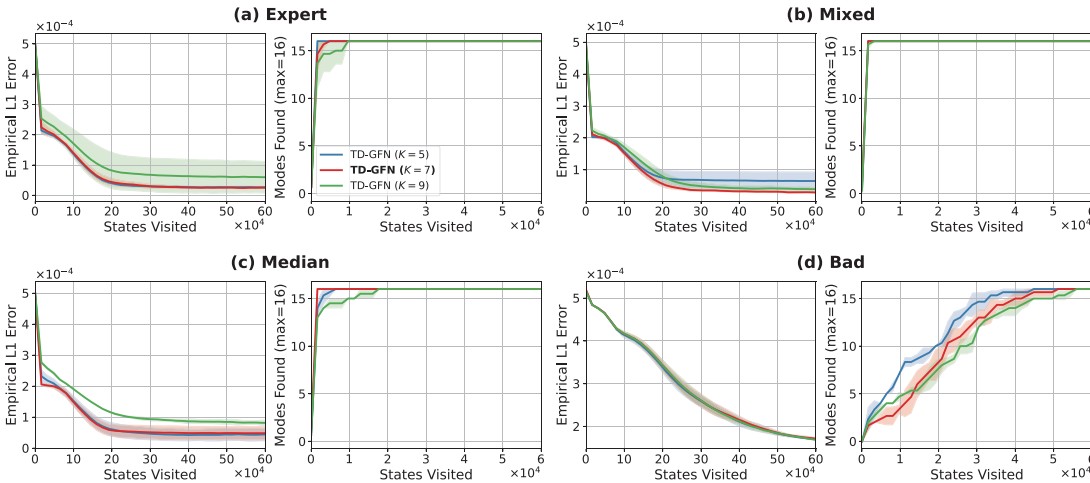

*Figure 12.* Performance comparison between different values of the pruning coefficient $K$ on the $8^4$ Hypergrid task. Policies are trained using four types of datasets, each consisting of $1,500$ trajectories.

### E.6. Time consumption and resource usage

For the Molecule Design task (Section 4.3), all algorithms are trained on an NVIDIA Tesla A100 80GB GPU. We report preprocessing time, per-epoch training time, total convergence time, and peak GPU memory usage for various GFlowNet training methods in Table 5.

For Proxy-GFN, preprocessing refers to training the proxy model. For COFlowNet, it involves constructing a dictionary that indexes all child nodes from the dataset for each state. For TD-GFN, preprocessing includes both learning the edge rewards and pruning the environment DAG, corresponding to Phase 1 and Phase 2 in Algorithm 1.

*Table 5.* Training time and GPU memory usage comparison across GFlowNet variants on the Molecule Design task.

| Method | Preprocessing Time | Epoch Time | Convergence Time | GPU Memory |
| --- | --- | --- | --- | --- |
| Proxy-GFN | 10h | 4.82s | 4h | 22.88G |
| Dataset-GFN | / | **0.56s** | 0.47h | 10.08G |
| FM-COFlowNet | 2h | 0.76s | 0.63h | 12.81G |
| QM-COFlowNet | 2h | 1.16s | 0.71h | 63.80G |
| TD-GFN (Ours) | **0.1h** | 0.59s | **0.20h** | 12.01G |

As shown, by predicting pruned edges through a single forward pass of a neural network—rather than querying from a pre-constructed dictionary—TD-GFN significantly reduces preprocessing time and incurs minimal computational overhead during training. Furthermore, owing to its rapid convergence (as demonstrated in Table 1), TD-GFN can learn a high-performing GFlowNet policy in under 30 minutes, highlighting its strong time efficiency.

Although the edge reward learning component introduces a modest computational cost, it remains negligible compared to the overhead of more complex mechanisms such as quantile matching. Despite this, TD-GFN outperforms methods enhanced with quantile-based techniques, underscoring its effectiveness and scalability.

### E.7. Ablation Study

In this section, we conduct ablation experiments on the $8^4$ Hypergrid task to evaluate the contribution of individual components within the proposed TD-GFN framework.

**Dataset Rebalancing.** As described in Section 3.1, we perform dataset rebalancing to adjust the offline data distribution to better approximate that of an expert policy. In Figure 13, we evaluate TD-GFN with and without rebalancing on both

the *Expert* and *Bad* datasets. For additional comparison, we also train the offline GFlowNet method COFlowNet on the rebalanced datasets.

The results show that rebalancing the *Expert* dataset does not degrade policy learning, despite the mild distortion introduced to the original data distribution. More importantly, rebalancing significantly improves policy performance when the dataset is collected under a behavior policy that diverges substantially from the expert.

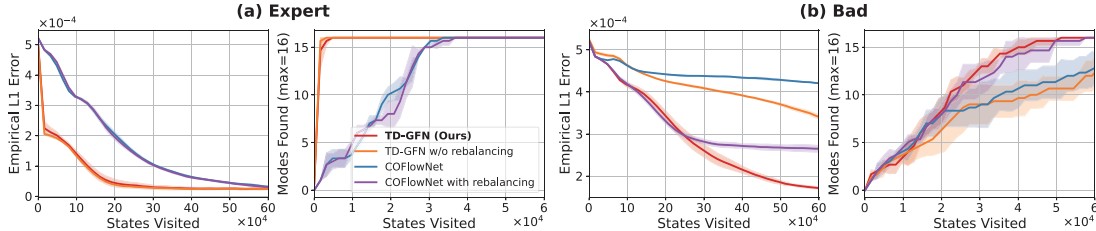

*Figure 13.* Performance comparison with and without dataset rebalancing on *Expert* and *Bad* datasets, each consisting of $1,500$ trajectories.

**Pruning and Backward Sampling.** To assess the individual impact of TD-GFN's components, we conduct ablations on the *Mixed* dataset. We compare the full TD-GFN with three variants: (i) a baseline without DAG pruning; (ii) a dataset-pruned variant that removes all edges not observed in the dataset; and (iii) a backward sampling baseline in which trajectories are sampled uniformly in reverse from terminal nodes, without using learned edge rewards. As shown in Figure 14, both the reward-guided pruning strategy and the prioritized backward sampling procedure are critical to the overall performance of TD-GFN.

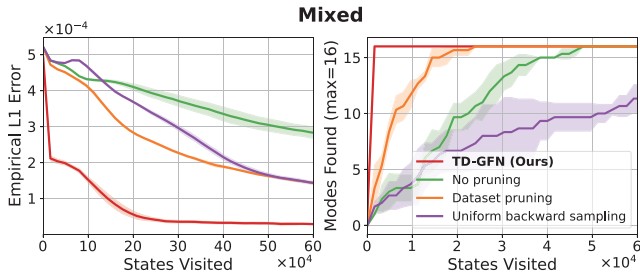

*Figure 14.* Ablation study on the *Mixed* dataset. We evaluate three TD-GFN variants: (i) no pruning; (ii) dataset-based pruning that removes unseen edges; and (iii) uniform backward sampling from terminal nodes. Each TD-GFN component proves essential for achieving optimal performance.

### E.8. Orthogonality to Training Paradigms

As shown in Algorithm 1, the improvements introduced by TD-GFN are orthogonal to the specific training paradigm employed after trajectory collection. Due to time and space constraints, our empirical analysis primarily focuses on the compatibility of TD-GFN with existing GFlowNet training objectives, as well as its integration with offline RL methods and the quantile matching technique proposed by Zhang et al. (2024).

**Compatibility with GFlowNet training objectives.** With trajectories sampled on the pruned DAG $G' = (V', E')$, we can readily apply various GFlowNet training objectives to optimize our policy on this subgraph without requiring substantial modifications, such as the flow matching (FM) objective (Bengio et al., 2021) and the trajectory balance (TB) objective (Malkin et al., 2022), as illustrated below.

By setting $R(s) = 0$ for all non-terminal states $s$, the FM objective for each sampled trajectory $\tau$ is given by:

$$\mathcal{L}_{FM}(\tau;\theta) = \sum_{s' \in \tau} \left( \sum_{s:(s \to s') \in E'} F_\theta(s, s') - R(s') - \sum_{s'':(s' \to s'') \in E'} F_\theta(s', s'') \right)^2, \tag{8}$$

where $F_\theta(s, s')$ denotes the flow along edge $(s \to s')$. The resulting flow-based policy is then defined by sampling child states in proportion to the learned flows at each parent state.

The TB objective for each sampled trajectory $\tau$ is given by:

$$\mathcal{L}_{TB}(\tau; \theta) = \left( Z_\theta + \sum_{(s \to s') \in E'} \left( \mathcal{P}_F^{G'}(s' \mid s; \theta) - \mathcal{P}_B^{G'}(s \mid s'; \theta) \right) - R(x) \right)^2, \tag{9}$$

where $\mathcal{P}_F^{G'}(s' \mid s; \theta)$ and $\mathcal{P}_B^{G'}(s \mid s'; \theta)$ denote the forward and backward transition probabilities respectively, both normalized over the edges in the pruned subgraph $G'$ is the reward of the terminal state $x$, and $Z_\theta$ is the learned normalizing constant. Moreover, we find it beneficial to fix $\mathcal{P}_B^{G'}$ to the form given in Equation (7), a practice also noted in Malkin et al. (2022). The results of applying the TB objective on the four datasets used in our main experiments are presented in Figure 15.

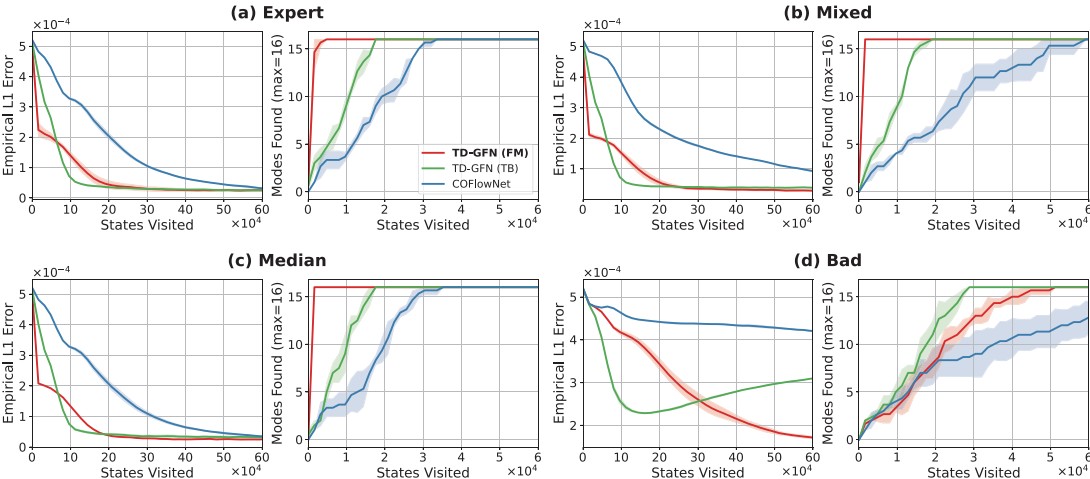

*Figure 15.* Performance comparison of TD-GFN with different training objectives as well as COFlowNet on the $8^4$ Hypergrid task. Policies are trained using four types of datasets, each consisting of $1,500$ trajectories.

We observe that the TB objective converges faster than the FM objective, but is less effective in rapidly exploring modes and exhibits weaker stability against overfitting on extreme datasets (e.g., *Bad*). We hypothesize this is because the flow function $F_\theta(s, s')$ in the FM objective can itself be interpreted as an implicit edge-level reward model. From this perspective, our pruning method effectively reshapes the learning landscape for this *flow-reward* model, simplifying its learning task, while our prioritized backward sampling guides it to strategically generalize to high-utility regions. Analogous to how a proxy reward model generalizes across terminal states, our approach enables this flow-reward model to generalize to unseen edges, thereby actively guiding the policy to explore novel states. This synergy might explain the FM objective's enhanced robustness and exploratory capability within our framework. Nevertheless, TD-GFN with the TB objective still substantially outperforms COFlowNet.

**Replacing GFlowNet objectives with Offline RL Training.** Instead of using the standard objectives in GFlowNet training, we examine the effect of replacing it with the Conservative Q-Learning (CQL) objective from offline reinforcement learning (Kumar et al., 2020), resulting in a variant we refer to as TD-CQL. We evaluate TD-CQL on both the *Expert* and *Mixed* datasets, each consisting of $1,500$ trajectories. Results are presented in Figure 16.

With training components inherited from TD-GFN, TD-CQL successfully discovers all modes using a considerably smaller number of state visits compared to CQL and achieves an *Empirical L1 Error* below $10^{-4}$ with fewer visits than TD-GFN, demonstrating strong sample efficiency and early-stage accuracy in modeling the target distribution. However, beyond this point, the *Empirical L1 Error* increases, indicating a decline in distributional approximation quality. This degradation arises from the nature of the offline RL objective, which tends to overfit the policy toward only the highest-reward objects, thereby limiting its ability to accurately model the full reward distribution.

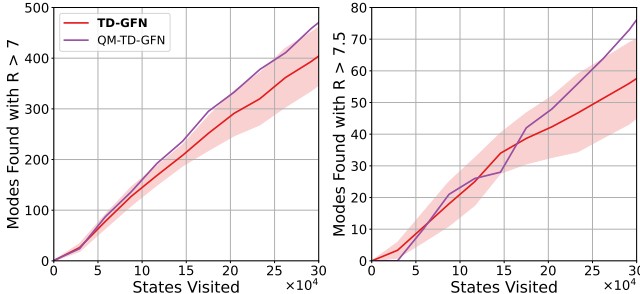

*Figure 16.* Performance of GFlowNet and CQL with and without our Trajectory-Distilled (TD) framework on the *Expert* and *Mixed* datasets, each consisting of $1,500$ trajectories.

**Incorporating Quantile Matching.**    Following the approach of Zhang et al. (2025b), we augment TD-GFN with the quantile matching technique proposed in Zhang et al. (2024), resulting in a variant referred to as QM-TD-GFN. We evaluate QM-TD-GFN on the Molecule Design task (Section 4.3) to assess its impact on the diversity of generated samples.

We observe that incorporating quantile matching into TD-GFN introduces significant training instability. This may be due to the pruned structure of the environment DAG, which reduces the need for a highly uncertain policy. Nevertheless, in cases where training remains stable, QM-TD-GFN is able to discover more high-reward modes than the original TD-GFN, as shown in Figure 17. These findings highlight the potential of quantile matching to enhance the diversity of generated molecules.

*Figure 17.* Number of high-reward modes discovered by QM-TD-GFN compared to TD-GFN.

## E.9. Discussion on Baseline Performance

To gain deeper insight into the behavior of baseline methods, we analyze results on the Hypergrid task, as shown in Figure 18.

Across datasets with varying levels of uncertainty, IQL excels at identifying high-reward modes. However, it also exhibits the highest *Empirical L1 Error*, underscoring a core characteristic of offline reinforcement learning algorithms: their objective often prioritizes recovering the most rewarding trajectories over accurately modeling the full reward distribution.

Although CQL is also an offline RL algorithm, its performance differs from IQL due to a distinct training paradigm. Unlike IQL, which learns exclusively from transitions in the dataset, CQL explicitly constrains policy actions to remain close to the data by penalizing the overestimation of Q-values. As a result, its performance is highly sensitive to dataset quality. A similar sensitivity is observed in COFlowNet, which applies conservative regularization to limit its policy coverage to dataset-adjacent regions.

In contrast, GAIL—when trained on a rebalanced dataset—demonstrates stable performance across datasets. It consistently maintains coverage over all modes and achieves an *Empirical L1 Error* slightly above $1 \times 10^{-4}$. Although GAIL performs slightly worse than Behavior Cloning (BC) in scenarios with abundant expert data, it significantly outperforms BC under noisy or sparse data conditions and achieves performance comparable to COFlowNet. These results highlight the benefit of incorporating reward signals from terminal states during training by rebalancing the dataset.

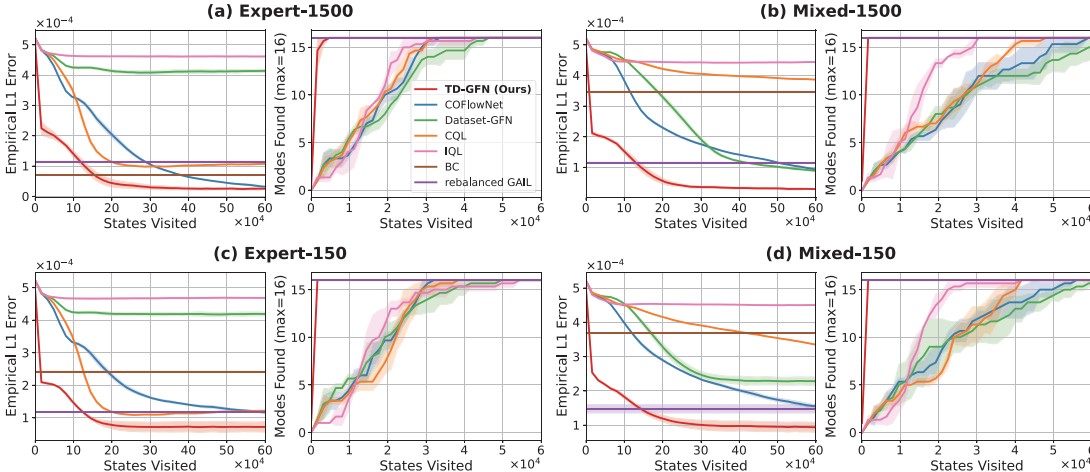

*Figure 18.* Performance comparison on the $8^4$ Hypergrid task. The top row uses $1,500$ trajectories from the *Expert* and *Mixed* datasets for policy training, while the bottom row uses only 150 trajectories.

### E.10. Analysis of Sample Diversity in Molecule Design

In the main paper (Figure 7), we report sample diversity using the number of high-reward modes discovered, which is a task-oriented metric evaluating the intersection of exploration and exploitation. To provide a more direct measure of structural diversity, we performed a post-hoc analysis by computing the average internal Tanimoto similarity among the top-100 highest-reward molecules sampled from the final policies of each method. A lower Tanimoto similarity score indicates higher sample diversity.

The results are presented in Table 6. As this table shows, our algorithm does not sacrifice diversity while ensuring sampling optimality (achieving the highest Top-100 Reward). This strong balance of high reward and low internal similarity explains why, as shown in Figure 7, our method is able to discover a significantly larger number of distinct high-reward modes.

*Table 6.* Comparison of Top-100 Reward (from Table 1) and Top-100 Internal Tanimoto Similarity (Avg. $\pm$ Std. over three seeds) on the Molecule Design task. Lower similarity indicates higher diversity.

| Method | Top-100 Internal Tanimoto Sim. ($\downarrow$) | Top-100 Reward ($\uparrow$) |
|---|---|---|
| Proxy-GFN | $0.665 \pm 0.024$ | $7.281 \pm 0.067$ |
| Oracle-GFN | $0.615 \pm 0.017$ | $\mathbf{7.408 \pm 0.021}$ |
| Dataset-GFN | $0.521 \pm 0.026$ | $7.198 \pm 0.018$ |
| FM-COFlowNet | $0.535 \pm 0.015$ | $7.201 \pm 0.015$ |
| QM-COFlowNet | $0.526 \pm 0.014$ | $7.296 \pm 0.022$ |
| **TD-GFN (Ours)** | $0.531 \pm 0.022$ | $\mathbf{7.450 \pm 0.037}$ |

### E.11. Naive Application of Conventional GFlowNet Objectives to Offline Datasets

To further investigate the necessity of our proposed framework, we evaluate the performance of standard GFlowNet training objectives—specifically Detailed Balance (DB) (Bengio et al., 2023), Trajectory Balance (TB) (Malkin et al., 2022) and Sub-Trajectory Balance (SubTB, with $\lambda = 0.9$) (Madan et al., 2023)—when directly applied to the offline trajectory datasets without additional guidance. Experiments were conducted on the $8^4$ Hypergrid task using both the Expert and Bad datasets ($1,500$ trajectories each).

As illustrated in Figure 19, these conventional methods perform poorly compared to TD-GFN. In the absence of structural guidance for unobserved regions, simply restricting standard online objectives to fixed offline trajectories leads to suboptimal convergence and limited mode discovery. These results underscore that the naive transfer of online GFlowNet objectives to the offline setting is insufficient, highlighting the necessity of the specific mechanisms introduced in our framework.

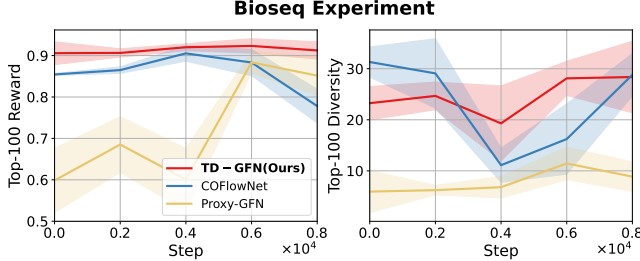

*Figure 19.* Performance comparison of conventional GFlowNet objectives when naively applied to $1,500$ offline trajectories in the *Expert* and *Bad* dataset on the $8^4$ Hypergrid task.

### E.12. Learning curves in Biosequence Design task

We provide the learning curves in Biosequence Design task in Figure 20. We omitted them in the main paper because, while TD-GFN remains stable throughout training, baseline methods exhibit significant volatility. To ensure a fair comparison, we reported the baseline results at the checkpoints that achieve the best balance between optimality and diversity.

**Bioseq Experiment**

*Figure 20.* Learning curves of three methods in the Biosequence Design task.

# F. Discussions

**Adaptability to Extended GFlowNet Settings.** TD-GFN can be directly applied to scenarios involving non-acyclic directed graphs (Brunswic et al., 2024), as it operates over general transition structures without a fundamental reliance on acyclicity. In contrast, adapting our framework to continuous environments would necessitate additional modeling considerations (e.g., for defining and pruning over continuous edge sets). Furthermore, extending the method to handle stochastic rewards or non-deterministic transitions (Pan et al., 2023c) represents an important direction, particularly for real-world applications where uncertainty is inherent in both reward evaluation and environmental dynamics.

**Toward More Advanced Edge Reward Modeling.** The current implementation of TD-GFN employs an adversarial IRL approach for edge reward estimation. This choice, while effective, inherits the well-known challenges of adversarial training, such as potential instability and sensitivity to hyperparameters. Therefore, the development of more stable IRL techniques presents a promising avenue for further enhancing the reliability and scalability of the TD-GFN framework. This may be achieved by leveraging recent advances in inverse reinforcement learning, such as stability-enhancing regularization (Fu et al., 2018; Kostrikov et al., 2020), representation learning–augmented IRL (Chandak et al., 2019), and score-based or energy-based IRL methods (Liu et al., 2020).

**On Dataset Generation.** Many of the datasets in our work are generated using GFlowNets. This strategy is also common in the offline reinforcement learning (RL) community (Fu et al., 2020; Gulcehre et al., 2020), where datasets are typically collected by RL policies of varying quality. However, our method is not tied to any particular data generation procedure. For instance, in Section 4.2, we employ a dataset curated from the DBAASP database (Pirtskhalava et al., 2021). We adopt GFlowNet-generated datasets not because our approach requires them, but because they provide a controlled and reproducible way to simulate realistic scenarios in which data arises from behavior policies of different optimality levels. This allows for a systematic investigation of how dataset quality influences performance. Moreover, certain real-world datasets were not included simply because no proxy-based learning baselines are available for comparison on them. In future work, we aim to further validate the effectiveness of our approach on datasets that more closely resemble practical deployment environments.

**What is Encoded in Trajectory Data.** Our work is situated in the offline GFlowNet setting, where training is conducted on a fixed collection of reward-labeled trajectories. These trajectories—whether collected from humans, heuristics, or simulators—are often overlooked in proxy-based methods. In contrast, our approach is designed to leverage such information without resorting to proxy reward models.

In this setting, however, our method differs from proxy-based approaches in that it explicitly leverages the trajectories themselves and assumes that they encode valuable information beyond terminal rewards. A natural question, then, is what constitutes a *valuable* trajectory. To deepen our understanding of the role of trajectories in offline GFlowNet training, we construct two additional datasets from the terminal states used in the main experiments by generating trajectories in reverse: (i) Uniform, where at each step a parent node is selected uniformly at random; and (ii) Rule-based, where the parent is chosen by preferentially reducing the dimension with the smallest current value, thereby producing trajectories that avoid the central regions of the hypergrid with low rewards. We then apply TD-GFN to these two datasets and compare the results with those obtained using the original GFlowNet-generated dataset. The results are presented in Figure 21.

Although all three datasets share the same set of terminal states, the results clearly show that the trajectories themselves provide crucial information. Trajectories generated via uniform backward sampling perform very poorly, and even the rule-based trajectories, which are highly efficient and deliberately avoid low-reward regions, are less effective than trajectories generated forward by a GFlowNet. This observation suggests TD-GFN leverages meaningful trajectory structure, implying it may struggle with datasets consisting only of noise. Nevertheless, its sensitivity is mild compared to COFlowNet, which degrades more significantly under such conditions.

**Future Directions** The observation above opens up two promising directions for future work. First, developing principled methods for generating effective trajectories from terminal states could extend TD-GFN's applicability to settings where only terminal data is available.

Second, the strong performance of TD-GFN on GFlowNet-generated datasets strongly suggests its potential in online and interactive training settings. We envision that the edge rewards could be learned and updated dynamically as the agent collects new experience. This dynamic guidance model could then be used to continuously improve sampling efficiency for

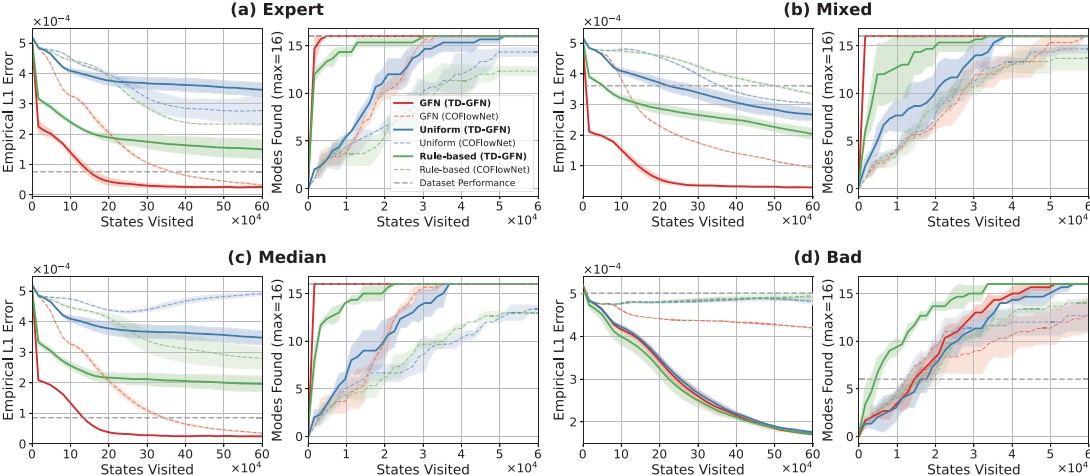

*Figure 21.* Performance comparison of TD-GFN policies trained on the $8^4$ Hypergrid task using three types of datasets with the same terminal states, each consisting of $1,500$ trajectories.

continued training, for instance, by intelligently focusing exploration on promising, high-reward regions of the DAG while steering the agent away from well-understood or low-utility pathways.

Furthermore, an interesting future direction would be to replace the "hard" pruning of the DAG with a "softer" guidance mechanism. In fact, the pruning method can be formally understood as an extreme case of a "soft" guidance mechanism, such as re-weighting the policy logits using the learned edge rewards. In this view, pruning is equivalent to applying an infinitely large positive weight to transitions above the threshold and a zero weight (or infinitely negative logit) to those below it. This perspective provides a clear theoretical motivation, as this re-weighting can be interpreted as imposing a strong Bayesian prior on the policy (incorporating the structural knowledge learned via IRL) or as a form of imitation learning regularization that constrains the policy to align with the distilled expert behaviors.

However, based on our initial explorations, while this "soft" method is a compelling alternative, it seems to be highly sensitive to its weighting-strength hyperparameter. A potential explanation is that it relies on the precise numerical accuracy of the learned edge rewards, which may contain generalization errors. Developing a more effective "soft" guidance mechanism remains a promising avenue for future research.

