# OpenReview forum: "Beyond the Proxy: Trajectory-Distilled Guidance for Offline GFlowNet Training"
_ICML.cc/2026/Conference — ICML 2026 regular_

### Official Review · Reviewer_KRi3 · 2026-03-12

**Soundness:** 4
**Presentation:** 3
**Significance:** 3
**Originality:** 4
**Overall Recommendation:** 5
**Confidence:** 4

**Summary:**

This paper proposes a novel algorithm for training GFlowNets using only an offline dataset. Existing methods rely on proxy reward models, which are prone to out-of-distribution generalization issues. The key idea is to incorporate an edge reward function that assigns value to intermediate state transitions based on their contribution to the terminal reward. This edge reward function is learned via an inverse reinforcement learning method (GAIL). Because the proposed method leverages the structural bias of high-reward trajectories (i.e., shared intermediate trajectories among high-reward terminal solutions), it enables better out-of-distribution generalization compared with methods that consider only terminal rewards, such as proxy-based approaches. The empirical study demonstrates the validity of the proposed method.

**Compliance With Llm Reviewing Policy:**

Affirmed.

**Final Justification:**

I gave a score of 5 after the authors’ response, as I believe this paper is novel and addresses an important problem for the GFlowNets community.

**Key Questions For Authors:**

1. Did the authors try other types of GFlowNet objective functions (e.g., SubTB)?

2. I am concerned that IRL could become a bottleneck for large-scale problems. Could the authors elaborate more on the stability of the IRL component?

3. Could this algorithm be applied orthogonally with proxy-based methods (i.e., is performance improved if online fine-tuning with a learned proxy is applied)?

**Limitations:**

None.

**Strengths And Weaknesses:**

**Strengths:**

1. This method is novel and technically sound. As far as I know, this work is the first application of inverse reinforcement learning to GFlowNet training.

2. The method cleverly leverages the DAG structure of GFlowNets to improve out-of-distribution generalization.

3. The method also appears to help with local credit assignment by using an edge reward function.

**Weaknesses:**

1. The method relies heavily on the performance of inverse reinforcement learning, which can also be unstable.

---

> ### Author Rebuttal · Authors · 2026-03-31
>
> Thank you for your time and effort in reviewing our paper! We are grateful for your constructive suggestions, which have significantly guided our improvements. Please find our responses to your comments below.
>
> ---
>
> **W1/Q2:** We appreciate this concern. A core design principle of TD-GFN is that IRL rewards are used *indirectly*—only for DAG pruning and prioritized backward sampling—while all gradient updates rely exclusively on ground-truth terminal rewards (Sec. 3.2). This significantly reduces the policy's sensitivity to IRL estimation errors.
>
> Moreover, the GAIL-based IRL framework we adopt has been theoretically and empirically demonstrated to learn reward functions with strong generalization [1, 2]. As shown in Sec. 3.1 and Appendix D.4, the learned edge rewards assign meaningful values even to transitions absent from the dataset, enabling the policy to generalize across the entire DAG rather than being restricted to observed regions.
>
> We further develop new theoretical analysis to formally characterize pruning robustness, which we will include in the revision. Using the paper's DAG notation, for a score function $g: E \to \mathbb{R}$ (the learned $R_E$) and threshold $\tau$, we remove edges with $g(e) < \tau$. Let $\mathcal{P}(x)$ denote the set of all directed paths from the root $s_0$ to terminal $x \in X$ in the original DAG. We define the *bottleneck score* $\beta_g(x) := \max_{P \in \mathcal{P}(x)} \min_{e \in P} g(e)$, which captures how well the best root-to-$x$ path withstands pruning.
>
> **Theorem** (Robust Preservation and Distortion Bound): Let $r^\*: E \to \mathbb{R}$ be the oracle edge score and $\hat{r}$ its learned estimate satisfying $\|\hat{r} - r^\*\|\_\infty \le \varepsilon$. Then (i) any terminal with oracle bottleneck margin $\beta_{r^\*}(x) - \tau > \varepsilon$ is guaranteed to survive pruning; (ii) among surviving terminals, relative probabilities under the reward-proportional target $p(x) \propto R(x)$ are *unchanged*—pruning only renormalizes; and (iii) the total variation distortion equals $\delta\_\tau = \mathbb{P}\_{x \sim p}[\beta\_{r^\*}(x) \le \tau + \varepsilon]$.
>
> **Corollary** (Mode Preservation): If disjoint mode basins $\mathcal{M}\_1, \dots, \mathcal{M}\_m \subseteq X$ each contain at least one terminal $x_i$ with $\beta_{r^\*}(x_i) > \tau + \varepsilon$, then no mode basin is eliminated by pruning.
>
> Catastrophic mode loss requires a stringent condition: *every* path from $s_0$ to *every* terminal in the basin must pass through at least one low-scoring edge below $\tau + \varepsilon$, which becomes increasingly unlikely as the DAG offers more paths. Moreover, surviving modes retain their exact relative probabilities. Combined with the $K$-sensitivity analysis (Appendix D.5) and strong empirical results across diverse conditions (Expert, Mixed, Bad, Few-shot in Sec. 4.1), IRL estimation noise does not directly propagate to policy quality.
>
> Regarding computational cost, as reported in Table 2, the IRL phase is already lightweight: TD-GFN's full preprocessing (IRL + pruning) takes only 0.1h on Molecule Design, versus 10h for proxy training and 2h for COFlowNet's dictionary construction. We discuss more stable IRL techniques in Appendix E.
>
> **Q1:** Yes. In Appendix D.8, we replaced FM with the trajectory balance (TB) objective in TD-GFN. TD-GFN+TB converges faster than TD-GFN+FM, while FM shows stronger mode exploration and robustness on extreme datasets (e.g., Bad); both substantially outperform COFlowNet across all datasets. Since TD-GFN's contributions (pruning + backward sampling) operate before the objective computation, they are architecturally orthogonal to the GFlowNet objective. We also evaluated SubTB, TB, and DB applied naively to offline data in Appendix D.11—these objectives perform poorly without TD-GFN's structural guidance, demonstrating the necessity of our framework.
>
> **Q3:** TD-GFN is designed to be orthogonal to downstream training paradigms. Appendix D.8 demonstrates compatibility with FM, TB, offline RL, and quantile matching. While we have not yet tested proxy-based online fine-tuning specifically, the architecture supports it naturally—the pruned DAG and prioritized sampling provide structural guidance independent of how reward signals are obtained downstream. In fact, as discussed in Appendix E, we envision that in an online setting, edge rewards could be dynamically updated as the agent collects new experience via a proxy, continuously improving sampling efficiency—we consider this a promising future direction.
>
> ---
>
> We hope these clarifications address your concerns. If so, we wonder if you could kindly consider raising your score? We will also be happy to answer any further questions you may have. Thank you very much!
>
> [1] Error Bounds of Imitating Policies and Environments for Reinforcement Learning, IEEE TPAMI 2021.
>
> [2] Reward-Consistent Dynamics Models are Strongly Generalizable for Offline Reinforcement Learning, ICLR 2024 spotlight.

---

> > ### Author Rebuttal · Reviewer_KRi3 · 2026-04-01
> >
> > My concerns have been resolved. I appreciate the new theoretical analysis provided by the authors, and I have increased my score accordingly.

---

> > > ### Author Response · Authors · 2026-04-07
> > >
> > > Thank you for your careful review and for raising your score to 5! We are glad that our responses have fully addressed your concerns, and we are grateful for the constructive feedback that pushed us to improve the work. Thank you again for your valuable feedback!

---

### Official Review · Reviewer_uXqf · 2026-03-13

**Soundness:** 3
**Presentation:** 3
**Significance:** 3
**Originality:** 3
**Overall Recommendation:** 5
**Confidence:** 4

**Summary:**

The paper studies the setting of training GFlowNets without proxy reward models, only relying on a trajectory-level dataset with rewards known only for a subset of terminal states. The authors propose TD-GFN algorithm. It learns edge-level rewards from the trajectory-level dataset by leveraging inverse reinforcement learning, and uses them to prune the environment graph and define a backward policy. After that, the GFlowNet is trained on both the trajectories from the initial dataset, and the trajectories sampled by the previously mentioned backward policy. An extensive experimental evaluation is provided on a number of tasks against baselines from both RL and GFlowNet literature.

**Compliance With Llm Reviewing Policy:**

Affirmed.

**Final Justification:**

The authors have addressed my concern regarding the lack of theoretical justification in the rebuttal.

**Key Questions For Authors:**

See Weaknesses.

**Limitations:**

No concern.

**Strengths And Weaknesses:**

**Strengths**:

1. An interesting setting of learning GFlowNets from offline data without relying on proxy reward models is studied in the paper, which has high practical potential in my opinion for future applications of GFlowNets.
2. The paper is fairly well-written and easy to follow.
3. The paper presents a diverse set of well-crafted experiments that shows clear evidence of the proposed method's utility and improved performance. The authors compare on a number of tasks against various GFlowNet and RL baselines, and use a broad range of metrics for comparison of the algorithms, including training convergence speed, distribution approximation quality, diversity and rewards of the generated samples. I also appreciated the detailed Appendix with extended experiments and ablations, as well as additional results on a real-world recommendation dataset.

**Weaknesses**:

The paper does not really present any theoretical justification or insights behind the proposed method. As it is presented in the text, to me it appears to be a set of carefully crafted heuristics. However, providing some level of theoretical analysis could really strengthen the paper in my opinion.

For example, the authors state that they "leverage theoretical equivalence between GFlowNet training and reinforcement learning". However, no analysis is presented on what is the edge-level reward that IRL learns, or why is this reward meaningful for pruning. An idea that came to me is that since GFlowNet-RL equivalence assumes using $\log P_B$ to define intermediate rewards (Equation 1), and the authors define a backward policy for sampling in their algorithm by normalizing the exponent of the reward learned by IRL (Equation 7), is it maybe possible to theoretically show that IRL actually recovers a specific backward policy? For example, assume that the trajectory dataset is sampled from a fixed GFlowNet (denote its forward policy $P_F$ and its backward policy $P_B$). Will the rewards that IRL recovers be close to $\log P_B$ if the dataset is sufficiently large? Or may they differ in an arbitrary way? Will the solution to the considered IRL problem be unique?

The same with pruning, even though the authors empirically demonstrate its benefits, no theoretical motivation or intuition behind it is presented.

Nevertheless, the paper presents solid contributions even in its current form, hence I recommend acceptance.

---

> ### Author Rebuttal · Authors · 2026-03-31
>
> Thank you for your time and effort in reviewing our paper! We are grateful for your constructive suggestions, which have significantly guided our improvements. Please find our responses to your comments below.
>
> ---
>
> **Response to Weaknesses:** We appreciate your insightful suggestion, especially the specific questions about the IRL-backward-policy connection. We agree the current submission does not make the theoretical underpinnings sufficiently explicit, and provide new formal analysis addressing both the IRL edge rewards and pruning below. We will include these results in the revision.
>
> **Setup.** Assume the trajectory dataset is generated by a fixed GFlowNet with forward policy $\mathcal{P}\_F^\*$, backward policy $\mathcal{P}\_B^\*$, and terminal reward $R(x)$. Let $F^\*(s)$ denote the state flow and $F^\*(s,s') = F^\*(s)\mathcal{P}\_F^\*(s'|s)$ the edge flow. We write $r^\*(s,s') := \log \mathcal{P}\_B^\*(s|s')$ for the canonical (oracle) edge reward, and $R_E(s,s')$ for the edge reward learned by IRL (Eq. 5).
>
> **What does IRL learn?** Under the parent-normalization gauge — requiring $\sum_{p \in \text{Pa}(s')} \exp(r(p,s')) = 1$ for every non-root $s'$ — the population IRL problem (Eq. 3) has a unique solution, which is exactly $r^\*$.
>
> *Theorem (Unique Recovery):* Among all rewards making $\mathcal{P}\_F^\*$ soft-optimal, the unique one satisfying parent normalization and terminal boundary $r(x, s_f) = \log R(x)$ (consistent with Eq. 1) is $r^\*(s,s') = \log \mathcal{P}\_B^\*(s|s')$.
>
> **Eq. (7) implements exactly this normalization**: it exponentiates $R_E$ and normalizes over parents, so it provably recovers the true expert backward policy. With sufficient data, this recovery is also consistent: if $\|R\_E - r^\*\|\_\infty \le \varepsilon\_N$ with high probability, then $\sup_{s'} \text{TV}(\hat{\mathcal{P}}\_B(\cdot|s'), \mathcal{P}\_B^\*(\cdot|s')) \le (e^{2\varepsilon_N} - 1)/2 \to 0$ as $N \to \infty$.
>
> Additionally, trajectory reweighting (Sec. 3.1) preserves $\mathcal{P}_B^\*$: if trajectories from a GFlowNet $(\mathcal{P}\_F, \mathcal{P}\_B, R)$ are reweighted by $w(x)$, the resulting distribution is a GFlowNet with the same $\mathcal{P}\_B$ and terminal weight $w(x)R(x)$, validating our rebalancing strategy.
>
> **Why is $R_E$ meaningful for pruning?** Since $r^\*(s,s') = \log(F^\*(s,s')/F^\*(s'))$, the canonical edge reward measures the fraction of expert flow reaching $s'$ that came through $s$. Low $R_E$ thus indicates transitions the expert rarely uses — providing a principled criterion for the pruning rule (Eq. 6).
>
> We formalize the robustness of this pruning via bottleneck analysis. For any terminal $x \in \mathcal{X}$, let $\mathcal{P}(x)$ denote all directed paths from root $s_0$ to $x$ in the original DAG. Define the *bottleneck score* $\beta(x) := \max_{P \in \mathcal{P}(x)} \min_{e \in P} r^\*(e)$, which captures how well the best root-to-$x$ path withstands pruning: a terminal with high $\beta(x)$ has at least one path whose every edge scores well, so it remains reachable even under aggressive thresholding.
>
> *Theorem (Robust Preservation and Distortion Bound):* Let $R_E$ be the learned edge reward with estimation error $\|R\_E - r^\*\|\_\infty \le \varepsilon$, and let $\tau$ be the pruning threshold (Eq. 6). Then: (i) any terminal with oracle bottleneck margin $\beta(x) - \tau > \varepsilon$ is guaranteed to survive pruning; (ii) among surviving terminals, relative probabilities under the reward-proportional target $p(x) \propto R(x)$ are *unchanged* — pruning only renormalizes; (iii) the total variation distortion satisfies $\|p - p_\tau\|\_{\text{TV}} = \delta\_\tau = \mathbb{P}\_{x \sim p}[\beta(x) \le \tau + \varepsilon]$.
>
> *Corollary (Mode Preservation):* If disjoint mode basins $\mathcal{M}\_1, \dots, \mathcal{M}\_m \subseteq \mathcal{X}$ each contain at least one terminal $x_i$ with $\beta(x_i) > \tau + \varepsilon$, then no mode basin is eliminated by pruning.
>
> Catastrophic mode loss requires *every* path to *every* terminal in a basin to pass through an edge with oracle score below $\tau + \varepsilon$ — a stringent condition that becomes increasingly hard to satisfy as the DAG offers more alternative paths.
>
> In our practical implementation (Eq. 6) , this threshold $\tau$ is parameterized by the hyperparameter $K$ such that $\tau = \text{mean}(\mathcal{D}\_{R_E}) - K \cdot \text{std}(\mathcal{D}\_{R\_E})$. Appendix D.5 empirically confirms this theoretical stability by demonstrating robust mode preservation without catastrophic loss across a wide range of $K$.
>
> ---
>
> We hope these clarifications address your concerns. If so, we wonder if you could kindly consider raising your score? We will also be happy to answer any further questions you may have. Thank you very much!

---

> > ### Author Rebuttal · Reviewer_uXqf · 2026-04-02
> >
> > I thank the authors for their detailed rebuttal and the provided theoretical findings. My concern regarding the lack of theoretical justification is addressed, so I would raise my score.

---

> > > ### Author Response · Authors · 2026-04-07
> > >
> > > Thank you for your careful review and for raising your score to 5! We are glad that our responses have fully addressed your concerns, and we are grateful for the constructive feedback that pushed us to improve the work. Thank you again for your valuable feedback!

---

### Official Review · Reviewer_DRXi · 2026-03-13

**Soundness:** 2
**Presentation:** 3
**Significance:** 2
**Originality:** 3
**Overall Recommendation:** 4
**Confidence:** 2

**Summary:**

This paper introduces TD-GFN, a proxy-free training framework designed to solve the challenge of learning Generative Flow Networks (GFlowNets) from static offline trajectory datasets. TD-GFN utilizes maximum causal entropy inverse reinforcement learning (IRL) on a rebalanced offline dataset (biasing trajectory sampling toward high-reward terminal states) to extract dense, transition-level edge rewards . The framework then applies reward-guided DAG pruning and prioritized backward sampling to transform these learned edge rewards into indirect structural guidance for the policy . To address the error propagation and out-of-distribution evaluation issues typical in standard proxy-based reward models, the authors ensure that gradient-based policy updates rely exclusively on recorded ground-truth terminal rewards rather than the IRL estimates . Additionally, the prioritized backward sampling mechanism strategically allocates the model's attention across terminal states and intermediate pathways to facilitate the effective exploration of unobserved high-utility regions. Experiments across synthetic Hypergrids, real-world biosequence design, molecule generation, and listwise recommendation scenarios demonstrate that TD-GFN significantly outperforms baselines like Proxy-GFN, COFlowNet, CQL, and IQL in convergence speed, distribution matching accuracy, and the discovery of diverse high-reward modes.

**Compliance With Llm Reviewing Policy:**

Affirmed.

**Ethical Review Concerns:**

The author's response largely addresses my query; however, it should be noted that the author needs to revise their phrasing, as their method does not rely solely on terminal rewards, but also requires trajectories in addition to the terminal rewards.

**Final Justification:**

The author's response largely addresses my query; however, it should be noted that the author needs to revise their phrasing, as their method does not rely solely on terminal rewards, but also requires trajectories in addition to the terminal rewards.

**Key Questions For Authors:**

1. The authors introduce a "Hard Pruning" mechanism to remove transitions with low edge rewards. However, this truncation could potentially sever critical paths leading to high-reward regions, especially since the rewards extracted via Inverse Reinforcement Learning (IRL) are inherently imperfect and subject to generalization errors. Could this hard thresholding lead to catastrophic mode dropping in highly complex environments with rugged reward landscapes?

2. In typical GFN applications such as molecular design, only terminal states and rewards are accessible in reality. The authors used a pre-trained GFN to generate synthetic trajectories to train TD-GFN. Therefore, in the molecular design scenario emphasized by the authors, since simulated trajectories must be used anyway, why not directly optimize the Proxy model itself?

**Limitations:**

1. Current leading offline GFN methods typically leverage offline data to construct experience replay buffers or integrate local search strategies (e.g., Local Search GFlowNets). The empirical study would be stronger with broader comparisons against additional recent offline GFlowNet methods, especially search-augmented or local-search-based variants tailored to offline settings.

2. The training pipeline of TD-GFN involves expensive upfront steps: Phase 1 requires running adversarial learning-based IRL (GAIL-style) to converge edge rewards; Phase 2 requires performing a global pruning evaluation on the DAG. However, the authors only focus on the number of trajectories required for updates during the final policy training stage (Phase 3). This makes it difficult to justify their claim of achieving 'high sample efficiency and convergence speed'.

**Strengths And Weaknesses:**

This paper on TD-GFN ingeniously integrates inverse reinforcement learning (IRL) with offline Generative Flow Networks (GFlowNets) to bypass the need for error-prone proxy reward models. By extracting innovative "edge rewards" and applying reward-guided DAG pruning, it effectively transforms static offline trajectories into dense, transition-level structural guidance. Furthermore, the utilization of prioritized backward sampling significantly enhances out-of-distribution mode discovery and accelerates convergence while isolating the policy from IRL approximation errors.

However, the study suffers from some theoretical and experimental flaws. These include a complete absence of rigorous theoretical bounds to verify that "hard pruning" avoids catastrophic mode dropping, the use of weak offline baselines lacking modern experience replay mechanisms, and the inappropriate omission of the massive upfront IRL computational costs in its core efficiency claims . Additionally, its reliance on proxy-simulated trajectories for core molecular design tasks creates a logical disconnect with its practical proxy-free motivation .

I am willing to raise my score if the authors can adequately address my concerns.

---

> ### Author Rebuttal · Authors · 2026-03-31
>
> Thank you for your time and effort in reviewing our paper! We are grateful for your constructive suggestions, which have significantly guided our improvements. Please find our responses to your comments below.
>
> ---
>
> Our responses to the Weaknesses are included in the answers below.
>
> **Q1:** The GAIL-based IRL we adopt has been theoretically and empirically shown to learn rewards with strong generalization [1, 2], assigning meaningful values even to unseen transitions (Sec. 3.1, Appendix D.4). Beyond this empirical reliability, we further prove that pruning is formally robust under bounded estimation error.
>
> For a score function $g: E \to \mathbb{R}$ (the learned $R_E$) and threshold $\tau$, we remove edges with $g(e) < \tau$. Let $\mathcal{P}(x)$ denote all directed paths from root $s_0$ to terminal $x \in X$. Define the *bottleneck score* $\beta_g(x) := \max_{P \in \mathcal{P}(x)} \min_{e \in P} g(e)$, capturing how well the best root-to-$x$ path withstands pruning.
>
> **Theorem** (Robust Preservation and Distortion Bound): Let $r^\*: E \to \mathbb{R}$ be the oracle edge score and $\hat{r}$ its learned estimate with $\|\hat{r} - r^\*\|\_\infty \le \varepsilon$. Then (i) any terminal with oracle bottleneck margin $\beta_{r^\*}(x) - \tau > \varepsilon$ is guaranteed to survive pruning; (ii) among surviving terminals, relative probabilities under the reward-proportional target $p(x) \propto R(x)$ are *unchanged* — pruning only renormalizes; and (iii) the total variation distortion equals $\delta\_\tau = \mathbb{P}\_{x \sim p}[\beta\_{r^\*}(x) \le \tau + \varepsilon]$.
>
> **Corollary** (Mode Preservation): If disjoint mode basins $\mathcal{M}\_1, \dots, \mathcal{M}\_m$ each contain at least one terminal $x_i$ with $\beta_{r^\*}(x_i) > \tau + \varepsilon$, then no mode basin is eliminated by pruning.
>
> Catastrophic mode loss requires *every* path to *every* terminal in a basin to pass through an edge scoring below $\tau + \varepsilon$ — a stringent condition that becomes increasingly hard to satisfy as the DAG offers more alternative paths. Our sensitivity analysis (Appendix D.5) confirms stability across a wide range of $K$ values, even on noisy datasets. We will include these results in the revision.
>
> **Q2:** Thank you for raising this point. The synthetic trajectories serve to *simulate* the offline scenario TD-GFN targets, not as a workaround. As stated in Section 5.2, "the training dataset simulates historical expert data" — reproducing the offline setting where trajectories exist but reward queries are expensive. The pre-trained GFN generates the offline dataset, after which the oracle is entirely inaccessible.
>
> Notably, TD-GFN matches or exceeds Oracle-GFN — which uses the ground-truth oracle as reward function — achieving Reward-10 of $7.733 \pm 0.036$ vs. $7.718 \pm 0.014$ with 16× faster convergence. While TD-GFN's performance depends on dataset quality (as with any offline method), this result demonstrates the effectiveness of trajectory-level structural guidance under the standard benchmark.
>
> **L1:** Thank you for the suggestion. We would like to clarify that Local Search GFlowNets [3] is an **off-policy** method — which still collects new trajectories and queries rewards online — rather than an **offline** method restricted to a fixed dataset. Its "Refining" step generates $\tau'$ via backtracking and reconstruction. It then applies filtering ($A(\tau, \tau') = \mathbf{1}\\{R(\tau') > R(\tau)\\}$ or Metropolis-Hastings with $R(\tau')/R(\tau)$) which requires evaluating $R(\tau')$ for each new trajectory — unavailable in the offline setting. To our knowledge, COFlowNet [4] is the most recent offline GFlowNet method, and we have included it as a primary baseline.
>
> **L2:** Appendix D.6 reports the full computational breakdown, where preprocessing refers to proxy training (Proxy-GFN), dictionary construction (COFlowNet), and IRL + pruning (TD-GFN):
>
> | Method | Preproc. | Convergence | Total |
> |---|---|---|---|
> | Proxy-GFN | 10h | 4h | ~14h |
> | COFlowNet | 2h | 0.63–0.71h | ~2.7h |
> | **TD-GFN** | **0.1h** | **0.20h** | **~0.3h** |
>
> TD-GFN's IRL preprocessing (0.1h) is **100× faster** than proxy training and **20× faster** than COFlowNet. Moreover, pruning uses a single neural network forward pass, adding only 0.03s/epoch (Dataset-GFN: 0.56s → TD-GFN: 0.59s, ~5% overhead). Total wall-clock: ~0.3h — an order of magnitude faster.
>
> ---
>
> We hope these clarifications address your concerns. If so, we wonder if you could kindly consider raising your score? We will also be happy to answer any further questions you may have. Thank you very much!
>
> [1] Error Bounds of Imitating Policies and Environments for Reinforcement Learning, IEEE TPAMI 2021.
>
> [2] Reward-Consistent Dynamics Models are Strongly Generalizable for Offline Reinforcement Learning, ICLR 2024 spotlight.
>
> [3] Local Search GFlowNets, ICLR 2024 spotlight.
>
> [4] COFlowNet: Conservative Constraints on Flows Enable High-Quality Candidate Generation, ICLR 2025.

---

> > ### Author Rebuttal · Reviewer_DRXi · 2026-04-03
> >
> > The authors have effectively addressed most of my concerns, but Q2 remains insufficiently resolved. In my view, this is not a minor experimental detail, but a central issue concerning the paper’s motivation and scope of applicability. The paper positions TD-GFN as a method for realistic settings where only terminal rewards are available, such as molecular design, and frames this motivation under the broader narrative of “Beyond the Proxy.” However, in the experimental setup, the offline data are still generated by a pretrained generative model to simulate historical trajectory data. As a result, the paper appears to demonstrate primarily that, when high-quality offline trajectories are already available, TD-GFN can leverage trajectory structure more effectively than existing methods, rather than fully establishing its necessity relative to proxy-based approaches in genuinely terminal-only settings. The authors’ response clarifies the construction of the benchmark, but it does not eliminate the tension between the paper’s motivation and its experimental setup.
> >
> > In other words, the paper does not yet sufficiently clarify, through experiments, whether TD-GFN can still maintain an advantage over proxy-based methods when trajectory information is weak, missing, or of low quality (i.e., when only terminal rewards are available), nor does it clearly delineate the boundary of that advantage. If the authors could provide a comparison that more closely reflects a terminal-only setting—for example, by retaining only terminal data and then constructing trajectories using a simple heuristic or other low-information procedure before applying TD-GFN, and comparing against proxy-based methods—I would be more inclined to raise my score.

---

> > > ### Author Response · Authors · 2026-04-03
> > >
> > > Thank you for the thoughtful follow-up. We appreciate the opportunity to further clarify the scope and motivation of TD-GFN.
> > >
> > > **Clarification of our setting.** We respectfully clarify that TD-GFN targets the setting of training GFlowNets from a **static dataset of trajectories with terminal rewards** — not from terminal states and rewards alone; it requires trajectory data as input. As discussed in our Introduction, execution trajectories in domains such as language modeling, recommendation systems (see also our real-world MovieLens experiment in Appendix D.1), and autonomous driving are inherently generated and readily available. This trajectory-level information is a cornerstone of offline reinforcement learning but often overlooked in proxy-based GFlowNet methods. The title "Beyond the Proxy" conveys that, given such trajectory data, TD-GFN avoids querying proxy models for out-of-distribution terminal rewards and thus eliminates error propagation introduced by proxy models — not that TD-GFN outperforms proxy-based methods when only terminal states and rewards are available. Accordingly, our comparison with Proxy-GFN and Oracle-GFN in Molecule Design is not meant to claim superiority over proxies in general. Rather, it shows that given a meaningful trajectory dataset, methods that leverage trajectory structure can outperform those that rely solely on terminal rewards to train a proxy.
> > >
> > > **Trajectory structure is essential — by design.** Given this setting, the natural follow-up — which you rightly raise — is whether TD-GFN requires trajectories to carry meaningful structural information. We fully agree: it does, and this is by design. We characterize this requirement both theoretically and empirically.
> > >
> > > *Theoretically*, we provide a new theorem:
> > >
> > > > **Theorem.** Suppose the offline trajectories are generated by a GFlowNet with backward policy $\mathcal{P}\_B^\*$. (i) Reweighting trajectories by a positive terminal factor $w(x_\tau)$ (e.g., $w(x)=R(x)$ in our Sec. 3.1 rebalancing) yields a distribution corresponding to a GFlowNet with the **same** backward policy $\mathcal{P}\_B^\*$. (ii) Among all solutions to the population max-causal-entropy IRL problem, the unique one satisfying the parent-normalization and terminal-reward boundary conditions is $r^\*(s,s') = \log \mathcal{P}\_B^\*(s|s')$.
> > >
> > > Part (i) ensures our rebalancing does not alter the backward policy IRL targets. Part (ii) shows the recovered edge reward — which corresponds to $\tilde{R}(s,s')$ in Eq. (1) — is precisely the log-backward-policy $\log \mathcal{P}\_B^\*$. Together: if trajectories are constructed via uniform random backward sampling, $\mathcal{P}\_B(s|s') = 1/\|\mathrm{Pa}(s')\|$, so the recovered reward $r^* = -\log|\mathrm{Pa}(s')|$ reflects only graph topology, carrying no reward-relevant guidance for pruning or backward sampling.
> > >
> > > *Empirically*, we investigate this in Appendix E ("What is Encoded in Trajectory Data," Figure 21). We construct two alternative datasets from the same terminal states by generating trajectories in reverse: (i) *Uniform*, selecting parents uniformly at random (using no reward information); (ii) *Rule-based*, a simple heuristic avoiding low-reward regions. As Figure 21 shows, uniform-sampled trajectories yield markedly worse performance, and even rule-based ones are less effective than those from a GFlowNet policy. This sheds light on the boundary you ask about: TD-GFN's advantage, like that of any offline method, improves with trajectory informativeness, and random construction does yield degraded results — consistent with our theorem. Notably, even under degraded conditions, TD-GFN's sensitivity remains milder than COFlowNet's.
> > >
> > > **TD-GFN does not require optimal trajectories to be effective.** While purely random trajectories are uninformative, the Molecule Design experiment shows TD-GFN succeeds with only moderate-quality data. We deliberately use a "moderately trained GFlowNet policy" (Behavior-GFN) to generate the dataset (Sec. 4.3). Methods that directly learn from it — BC (Reward-100: $7.223$), GAIL ($7.152$), Dataset-GFN ($7.198$) — achieve performance comparable to Behavior-GFN itself ($7.220$), unable to substantially exceed it, confirming the trajectory information is insufficient for naive methods to fully exploit. In contrast, TD-GFN achieves $7.450$, significantly surpassing Behavior-GFN. This demonstrates that TD-GFN's gains arise not from imitating the trajectory distribution, but from its out-of-distribution generalization mechanisms — IRL-guided pruning and prioritized backward sampling — which extract structural information that simpler methods fail to utilize.
> > >
> > > We will revise the paper to foreground the discussion of trajectory structure information (currently in Appendix E) and more explicitly position our setting. We hope this clarifies the scope of TD-GFN's applicability, and we would be very grateful if you could consider re-evaluating your score in light of these clarifications.

---

### Decision · Program_Chairs · 2026-04-30

**Decision:**

Accept (regular)

**Comment:**

This paper proposes an algorithm to improve offline GFlowNet training using inverse reinforcement learning to provide an edge-level reward. All of the reviewers, some of whom are experts on GFlowNet algorithms, found the idea interesting and potentially useful in practical applications where exact rewards are not available for all terminal states, relevant to biological and chemical structure discovery tasks where GFlowNets have been applied. There were initially concerns about lack of theoretical justification, which were resolved by new analysis that satisfied Reviewers KRi3 and uXqf.

One of the reviewers raised an ethics concern about some of the phrasing; the authors made clarifications in a note to the AC and I believe this concern is not relevant.

Thus, the paper was assessed positively overall and no significant unaddressed weaknesses remain.

I hope the rebuttal discussion will be taken into account for the final revision. I would also add to the reviewers' comments that the claim of "superior fit to the target distribution" in §1 is not supported by the experiments: the chosen metrics of empirical L1, cumulative number of modes found, and top-k reward are not measures of fit to the target distribution, which is the problem GFlowNets aim to solve N.B., cumulative modes found can often be increased simply mixing in a uniform policy). As the aim of the paper seems to be diverse mode discovery and not distribution matching in itself, this claim needs to be revised.